# Creep-type all-solid-state cathode achieving long life

Xiaolin Xiong ®[1,2], Ting Lin ®[1], Chunxi Tian[1,2], Guoliang Jiang[1,2], Rong Xu ®[3] ✉,
Hong Li ®[1], Liquan Chen[1] & Liumin Suo ®[1,2] ✉

Electrochemical-mechanical coupling poses enormous challenges to the interfacial and structural stability but create new opportunities to design innovative all-solid-state batteries from scratch. Relying on the solid-solid constraint in the space-limited domain structure, we propose to exploit the lithiation-induced stress to drive the active materials creep, thereby improving the structural integrity. For demonstration, we fabricate the creep-type all-solid-state cathode using creepable Se material and an all-in-one rigid ionic/electronic conducting $Mo_6Se_8$ framework. As indicated by the in-situ experiment and numerical simulation, this cathode presents unique capabilities in improving interparticle contact and avoiding particle fracture, leading to its superior electrochemical performance, including a superior long-cycle life of more than 3000 cycles at 0.5 C and a high volumetric energy density of 2460 Wh/L at the cathode level. We believe this innovative strategy to utilize mechanics to boost the electrochemical performance could shed light on the future design of all-solid-state batteries for practical applications.

Electrochemical-mechanical failure at interfaces is crucial for all-solid-state batteries (ASSBs). Unlike the flowable liquid electrolyte (LE) that can accommodate the volume change of active materials (AM) to guarantee the solid-liquid conformal interface and dissipate internal stress, the solid electrolyte (SE) usually displays a rigid character, and thus, it has poor interfacial stability susceptible to the volume change of active particles during (de)lithiation due to being squeezed with active particles in a constrained electrode space[1–3].

Detailly, as Fig. 1 shows, the rigid SE (oxide-based SE) and the rigid AM (transition metal oxide) usually form the hard point-point contact inside the electrodes, resulting in a large interfacial resistance that severely impedes the ion transport and lowers the electrode kinetics[4,5]. Meanwhile, for rigid AM-rigid SE interfaces (type I), the accumulated stress with cycling leads to the detachment of interfacial contact, fragmentation, and, ultimately, the electrochemical death of brittle particle particles[6]. If the relatively soft sulfide-based SE is instead of the rigid oxide-based SE, the soft SE and the rigid AM particles (type II) can

develop face-to-face initial contact due to the SE deformation under the external stack pressure. However, as the deformation of sulfide SE may not be fully elastic, it fails to adjust dynamically to the expansion and contraction of active particles during (de)lithiation; thus, the interfacial detachment stands inevitable with the time-dependent cycling[7].

With the aforementioned challenges, it is not viable to duplicate the configuration of the liquid lithium-ion batteries to address the electrochemical-mechanical dilemma in ASSBs. Hence, electrode materials with appropriate electrochemical-mechanical characteristics are necessary for the ab initio design of well-performing all-solid-state electrodes. Given the distinct chemomechanics in ASSBs, we are inspired to exploit the electrochemically induced stress to drive the time-dependent deformation of creepable active materials to form conformal interfaces. Such a transition from passive accommodation to active adoption holds the promise to fundamentally resolve the electrochemical-mechanical issues.

[1]Beijing National Laboratory for Condensed Matter Physics, Institute of Physics, Chinese Academy of Science, Beijing 100190, China. [2]Center of Materials Science and Optoelectronics Engineering, University of Chinese Academy of Sciences, Beijing 100049, China. [3]State Key Lab for Strength and Vibration of Mechanical Structures, Department of Engineering Mechanics, Xi'an Jiaotong University, Xi'an 710049, China. ✉e-mail: rongxu@xjtu.edu.cn; suoliumin@iphy.ac.cn

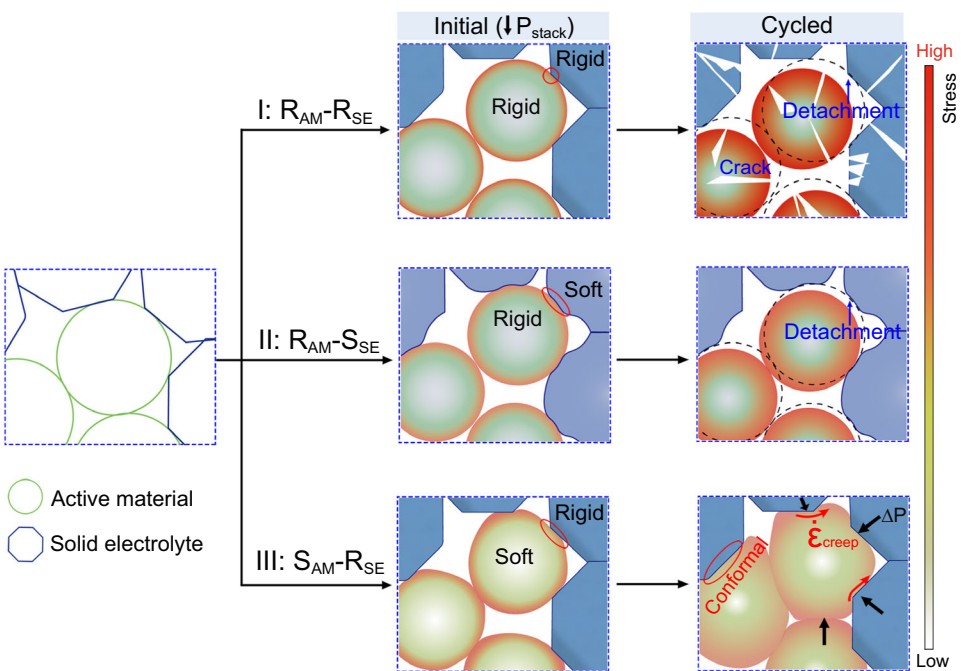

**Fig. 1 | Interfacial evolution inside all-solid-state cathodes.** Evolutionary models of all-solid-state electrode stability with different components over time: type I, rigid active material-rigid solid electrolyte; type II, rigid active material-soft solid electrolyte; type III, soft active material with time-dependent deformation and rigid solid electrolyte.

## Results

### Design and Construct Creep-type all-solid-state cathodes

In classical mechanics, materials are commonly characterized as brittle and ductile according to their capability to deform plastically, as represented in the deformation mechanism maps in the span of the shear modulus ($\tau/G$ = shear stress/ shear modulus) against the homologous temperature ($T_H = T/T_m$ = working temperature/melting temperature) (Fig. 2a). Materials with a low homologous temperature ($T_H < 0.3$), such as the transition metal oxides[4], are brittle and susceptible to fracture under stress. Whereas materials with an elevated homologous temperature ($T_H > 0.3$), such as the alloy-type electrode materials (Se, Te, Sn, etc.), can change shape via self-diffusion under shear stress, triggering a polycrystalline solid to behave like a viscous fluid at the macroscopic level (Fig. 2a, Table S1, S2)[8–11]. This phenomenon is known as creep, which typically refers to the time-dependent component of plastic deformation under long-term exposure to the stress gradient[12,13]. So far, creep engineering has been applied to avoid stress concentration or void formation in the single-phase Li/Na metal anode[14–18]. However, the creep-type all-solid-state cathode has not been achieved due to the lack of the creep occurrence environment in the multi-phase porous architectures of the cathode.

Considering the creep is the propensity of solid material with high $T_H$ to deform under long-term exposure to mechanical stress, inspiringly, we propose designing creep-type all-solid-state (CT-ASS) cathodes to address their electrochemical-mechanical issues, intelligently leveraging stress-driven creep of active materials to improve interfacial contact and continuously release lithiation stress. The creepable alloy cathode material is our first choice to prove our concept of CT-ASS electrodes. Foremost, a deformation-constraint space with a rigid skeleton must be created, ensuring that the expansion of the creepable alloy active material during lithiation is restricted for generating the corresponding stress gradient. Moreover, the CT-ASS cathodes should have proper porosity where the stress generated from the volume change of alloy particles can exert force onto the alloy itself to induce material creep to the pore space to release the stress. In this context, time-dependent creep inside the cathodes can intelligently

and automatically improve interfacial contact with cycling and reduce internal stress.

Among all the alloy cathode candidates, the creepable Se well balances all comprehensive merits, including the high volumetric capacity (3253 mAh/cm³) and much higher electronic conductivity (Se: $10^{-3}$ S/cm)[19] (Fig. 2b, Table S2). This indicates that, with an approximate volumetric capacity, Se can occupy a larger active content within the electrode than S (3467 mAh/cm³, $5 \times 10^{-28}$ S/cm)[20], indicating that it is the promising active materials for CT-ASS cathode. However, previous traditional all-solid-state Se electrodes incorporate a large amount of solid electrolyte (SE) and high specific surface area conductive carbon (>10 wt.%), resulting in the electrode with a porous ion (SE) - electron (C) conducting skeleton, which is compressible by the volume change of active particles to weaken the mutual confinement between particles. Thus, it is impossible to induce substantial stress on the Se particles to drive their creep and improve the electrodes' structural integrity in traditional all-solid-state electrode (Fig. S2). Therefore we must construct the electrode with a rigid framework to develop substantial stress in the constraint space to drive the Se creep.

As a prototype, we design an all-electrochemical-active all-solid-state Se cathode by employing all-in-one mixing ionic/electronic Chevrel phase Mo₆Se₈ with the high density ($D_{Li}$:$10^{-8}$–$10^{-10}$ cm²/s; $\sigma_e$: 32.4–154.2 S/cm; $\rho = 6.52$ g/cm³, Figs. S3–S6, Table S3)[21] to construct the rigid self-constraining framework (Fig. 2c, Fig. S7). It shows that the 40 wt.% Se+60 wt.% Mo₆Se₈ (Se-60MSe, mixing ratio with the best performance, Figs. S7, S8, Table S4) cathode has much lower porosity and higher density than the conventional 40 wt.% Se+50 wt.% Li₆PS₅Cl (LPSC)+10 wt.% Super P (Se-50SE-10SP) cathode in the range of applied stack pressure from 100 to 360 MPa (Fig. 2d, Fig. S9, Table S4, S5).

### Creep behavior of Se inside the all-solid-state cathode

Subsequently, to investigate the creep behavior of Se during cycling, we employ an in-situ SEM apparatus (Fig. 3a, b) to track the morphological evolution of the Se-60MSe cathode (the complete process is shown in Supplementary Movie 1). Figure 3c shows the electrode morphology at the initial state (2.3 V), the end of discharge (1.4 V), and

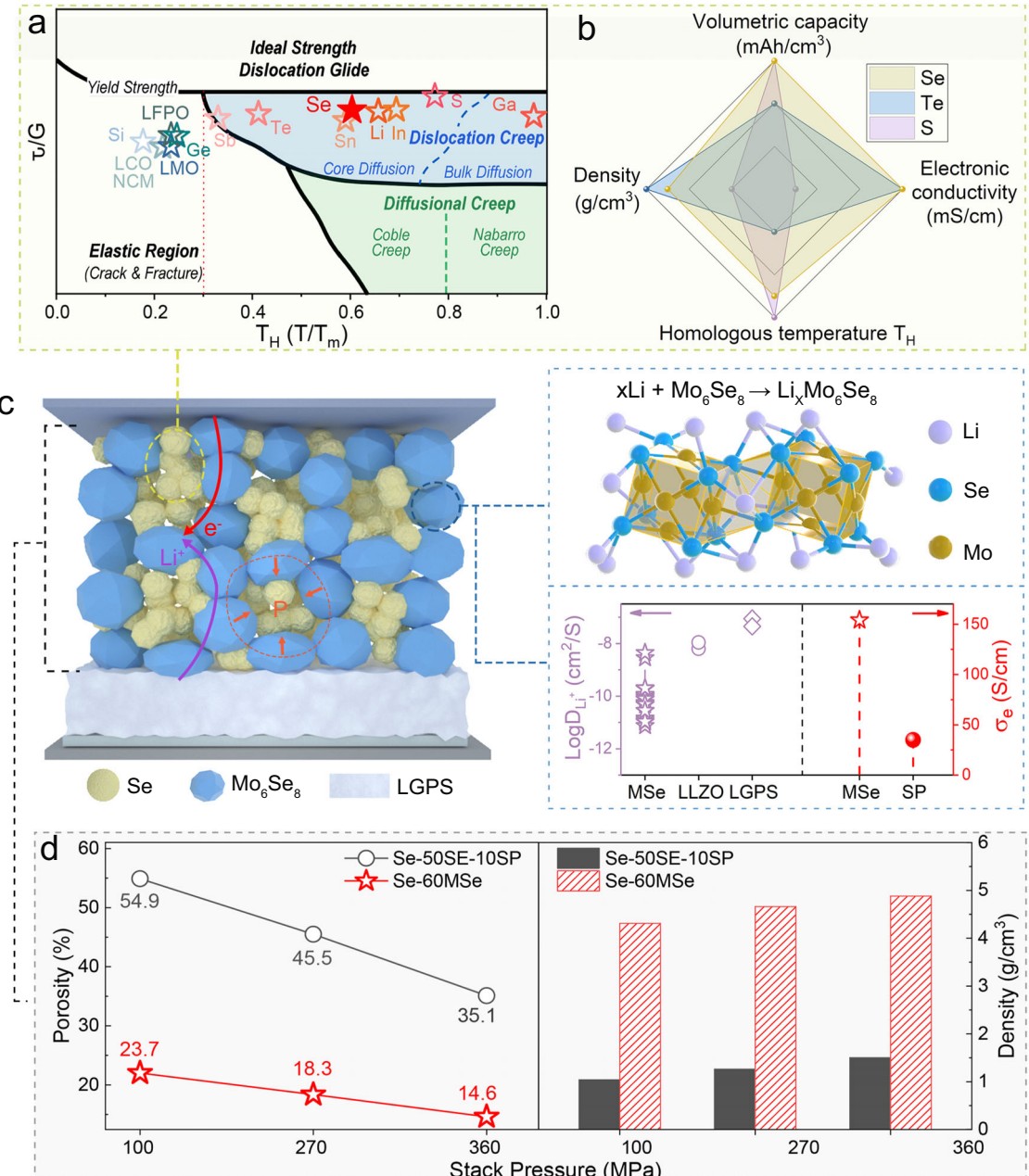

**Fig. 2 | Construct CT-ASS cathodes. a** Deformation mechanism maps for materials at different stress states and temperatures ($\tau$: shear stress, $G$: shear modulus)[40-45]. Particles inside ASSBs are subjected to shear stress greater than the external stack pressure (Fig. S1)[5]. **b** Comparison of the performance metrics of common cathode materials[19,46]. **c** The structure diagram of the creep-type Se-60MSe cathode. The inset shows the ionic and electronic conductivity of MSe[47,48]. **d** Comparison of the porosity and density of Se-50SE-10SP and Se-60MSe all-solid-state cathodes.

the end of charge (3.4 V), respectively. As shown in Fig. 3c-II, III, the pores undergo a distinct shrinkage with an evident enhancement in interparticle contact during discharge (lithiation), and the Se particles retain their structural integrity without being pulverized or broken. During subsequent charge (delithiation), Li-ion is extracted from Se particles, theoretically inducing Se volume retraction and pores enlargement. Supposedly, if there is no irreversible creep deformation, the pores volume should re-expand to 78% of the initial volume (Fig. S10), because the degree of lithiation of Se remains at 22% (calculated by the Columbic efficiency (CE)) at the end of charge (Fig. 3c-V). However, the actual porosity further contracts during charge (delithiation, Fig. 3c-IV, V), implying that the particles continue to deform into the pores due to Se creep. This phenomenon can be exclusively ascribed to the creep deformation induced by the internal

stress exerted on the Se particles by the rigid $Mo_6Se_8$ framework. We calculate the relative areas of individual 20 pores to substantiate our observations further. The results reveal that most pore sizes are much smaller than 78% of the initial state, and eight pores undergo further shrinkage during delithiation (Fig. 3d, Fig. S11), highlighting the pervasive nature of Se/Li₂Se creep within the electrode structure. This is a direct observation that Se creeps throughout the charging and discharging process, forming improved contact and leasing stress in the electrode.

Since it is challenging to characterize the dynamic evolution of stress and strain inside the cathode by existing experimental technique, we perform numerical analysis to facilitate a deeper understanding of the (de)lithiation-stress-creep coupled behavior of Se in the all-solid-state cathode during cycling. We construct a

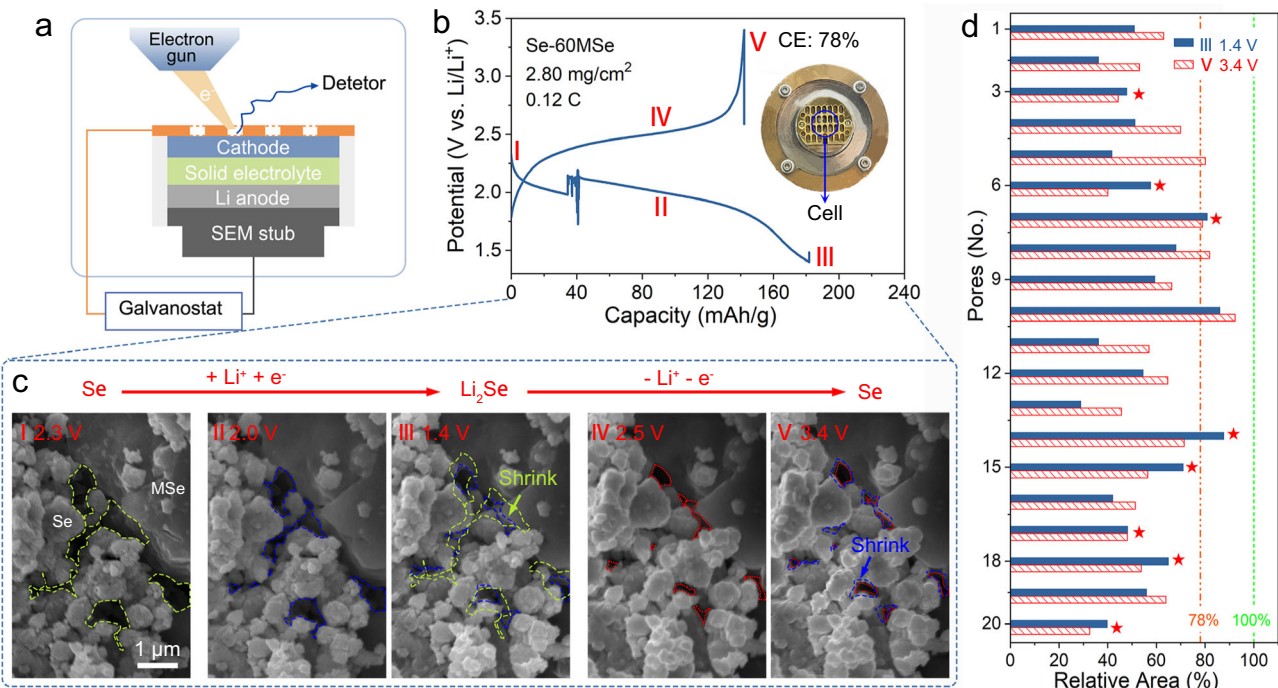

**Fig. 3 | In-situ observation of Se creep evolution during galvanostatic discharging-charging. a** Schematic illustration of the Se-60MSe|LPSC|InLi cell for in-situ scanning electron microscopy (SEM) observation during the electrochemical reaction. **b** The corresponding galvanostatic discharge-charge curves. The minor deviation in the discharge curve is due to the device's instability in the inset. **c** The morphology of the Se-60MSe cathode varies with different states of charge, (I) the initial state−2.3 V, (II) 2 V, (III) the end of discharge−1.4 V, (IV) 2.5 V, (V) the end of charge−3.4 V. Dashed lines mark the pore contours. **d** Comparison of the relative area of pores.

representative volume element (RVE) in which the volume fractions of Se, MSe, and porosity (35.6 vol.%, 39.7 vol.% and 24.7 vol.%, respectively) are adjusted to be consistent with the CT-ASS Se-60MSe cathode in experiment (Fig. 4a). The fields of deformation and stress (i.e., equivalent stress) in the cathode at its fully lithiated state are resolved in the Fig. 4b. Along with the volume expansion during lithiation, the Se particles tend to form the conformal contact with other Se and MSe particles. The stress developed in the cathode remains relatively low at around 0.2 GPa. For comparison, we build a control cathode with the non-creepable active material LiCoO$_2$ (LCO). During its lithiation, most LCO-LCO and LCO-MSe contacts are presented as the point contact, and the developed stress can reach as high as ~3.0 GPa (Fig. S12). This significant difference in contact condition and stress evolution is attributed to the time-dependent creep behavior of Se (Fig. S13). In specific, the volume expansion of Se during its lithiation incurs stress inside the cathode, which in turn facilitates the Se particles to deform toward the stress-free pore via creep. Upon the long-term lithiation, the creep strain accumulates (Fig. 4c, Fig. S14), driving the Se particles to conform with surrounding particles, during which the elastic strain and corresponding stress are relaxed.

Additionally, we quantify the contact condition (Fig. 4d) and stress level (Fig. 4e) in the CT-ASS Se-MSe cathode and LCO-MSe cathode upon their cycling at a rate of 0.1 C. Each cycle includes a lithiation process and subsequent delithiation process. Herein, we define the particle contact ratio as the contacted surface area of particles divided by their total surface area. The LCO-LCO and LCO-MSe contact ratios in the fully lithiated LCO-MSe cathode are 3.84% and 9.10%, respectively (Fig. 4d). Since the LCO and MSe are assumed to be linear elastic materials without any creep behavior, the particle contact ratios remain constant throughout the entire cycling. In contrast, the Se-Se and Se-MSe contact ratios in the Se-MSe cathode can reach 70.5% and 67.0% in the first cycle and gradually increase to 84.4% and 79.5% in the 20$^{th}$ cycle, which adds effective electrochemical site and increases

the continual electron and ion paths inside the cathode. From the structural integrity perspective, the average equivalent stress developed in the Se and MSe particles inside the Se-MSe cathode is only 0.099 GPa and 0.171 GPa, respectively, and further drops to 0.067 GPa and 0.156 GPa at the 20$^{th}$ cycle. This reduced stress level inside the cathode caused by the accumulated Se creep benefits the structural integrity of the cathode and its long-term electrochemical performance. In contrast, the stress developed in the LCO-MSe cathode climbs to the GPa level (Fig. 4f, Fig. S15), which inevitably incurs electrode breakdown and the corresponding performance decay[22].

To further verify the effect of the creep of Se on the structural integrity and electrochemical performance of all-solid-state cathodes, we evaluate the cycling performance of the CT-ASS Se-60MSe cathode at 0.05 C (Fig. 5a). It presents superior cycling stability with high-capacity retention of 97.2% after 3200 h cycling (Fig. 5a). Interestingly, the CT-ASS Se-60MSe cathode shows the continuously increasing electrochemical-active utilization of Se from 76.8 ca.% (518.2 mAh/g) to 90.1 ca.% (608.2 mAh/g) in the initial cycles, indicating that more Se particles are evolved in the electrochemical reaction with cycles, mainly due to the improved inter-particle contact by time-dependent Se creep, as demonstrated by in-situ SEM observation and numerical analysis.

Next, we chose the non-creepable FeS ($T_{H,FeS}$ = 0.23) as the control material to the creepable Se ($T_{H,Se}$ = 0.69, Table S2), as they have similar capacity (Se, 675 mAh/g; FeS, 609 mAh/g) and volume change (Se, 98%; FeS, 100%)[23,24]. We first evaluate the mechanical properties of different materials at room temperature, including Young's modulus, hardness, and strain-rate sensitivity exponent (i.e., the sensitivity of the creep rate to the stress)[25] by the nanoindentation in a specialized fluid cell where the sample was immersed and protected in mineral oil[26] (Fig. 5b, Figs. S17–S26). The samples used for nanoindentation tests have a dense and smooth surface, and the particles intimately contact with each other without invisible pores and cracks, thus the

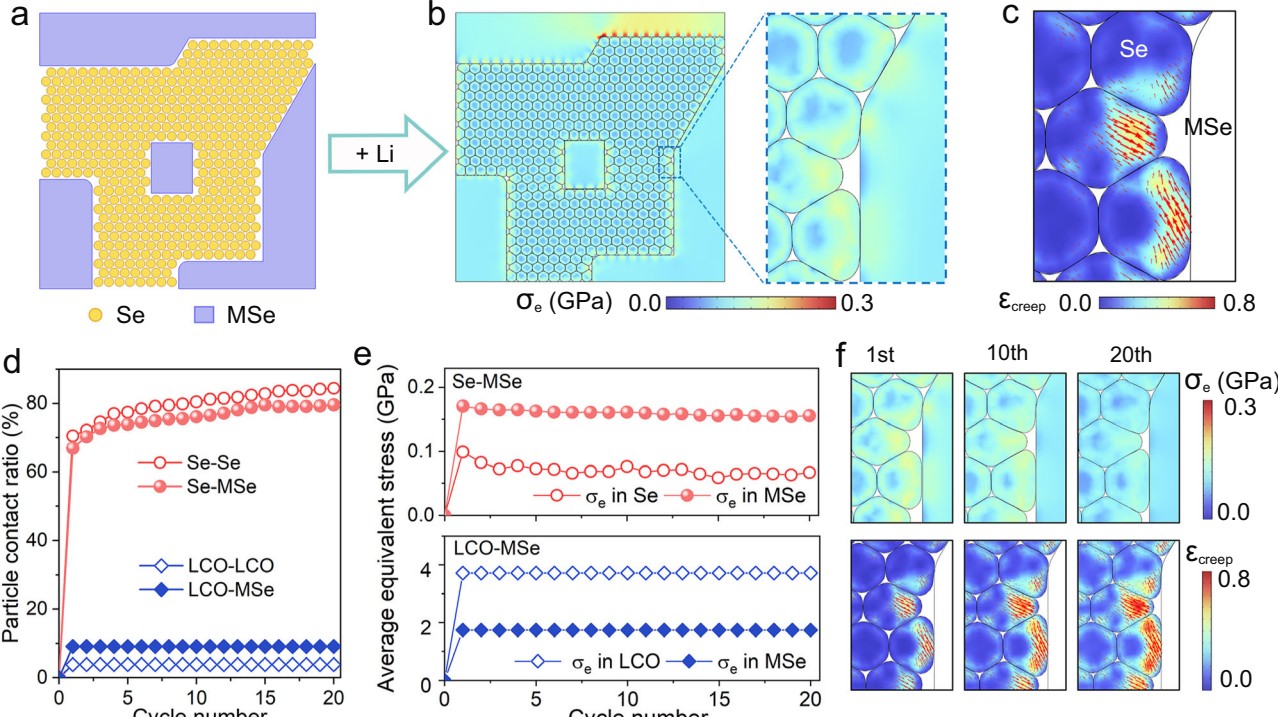

**Fig. 4 | Numerical analysis of the creep behavior of the all-solid-state cathodes. a** The RVE for the Se-MSe cathode. **b** The deformation and equivalent stress in the fully lithiated Se-MSe cathode. **c** The creep strain's magnitude (color contour) and direction (arrows) in the Se particles. **d** The particle contact ratio and **e** average equivalent stress in the Se-MSe and LCO-MSe cathodes upon galvanostatic cycling. **f** The equivalent stress and creep strain in the CT-ASS Se-MSe cathode at different cycles. The parameters for numerical analysis are listed in Table S6.

mechanical properties measured from such dense pellets should be close to the intrinsic ones of the materials (Fig. S19). As shown in Fig. 5b, Se exhibits a higher strain-rate sensitivity exponent (comparable to Li metal)[14,27], demonstrating that Se is more susceptible to creep than FeS. In addition, the nanoindentation test holding peak load for over 100 s of Se material further indicates that the Se is prone to continual creep (Figs. S27, 28). Furthermore, the lithiated $Li_2Se$ exhibits a higher strain-rate sensitivity exponent with considerable stiffness and Young's modulus reduction (Figs. S22, S23), illustrating that the $Se/Li_2Se$ material progressively creeps throughout the charging and discharging process, which is consistent with the in-situ SEM observation. Accordingly, the electrodes with Se (Se-60MSe) and FeS (FeS-60MSe) as active materials show substantial differences in their cycling performance and microstructural evolution. Similar to the results in Fig. 5a, the Se-60MSe cathode presents an increase in available capacity until 500 cycles (Fig. 5c) due to the gradually improved contact between particles inside the cathode and maintains superior cycling stability with a capacity retention of 81.8% after 1000 cycles. However, the capacity of the FeS-60MSe cathode fades monotonically with a capacity retention of only 35% after 1000 cycles.

Moreover, the Se-60MSe cathode presents a tiny change in thickness (4.76%) and a pore-free surface with a denser particle pack than its initial state (Fig. 5d, e). Conversely, the thickness of the FeS-60MSe cathode is significantly increased (66.7%), and the particles experience severe breakage and pulverization (Fig. 5f, g), as predicted by the numerical analysis that the accumulated stress would eventually cause the particles to fragment in the non-creepable all-solid-state cathode. In conclusion, although Se and FeS have similar capacity and volume change, the Se-60MSe cathode with creepable Se demonstrates superior cycling stability and structural integrity than the FeS-MSe cathode with non-creepable FeS, mainly due to the improvement of particle contact and stress relaxation by the Se creep.

## Superior electrochemical performance of CT-ASS cathodes

To evaluate the electrochemical performance of the CT-ASS cathodes in practical applications, we assemble the ASSB (Se-60MSe|LPSC| In$_x$Li) using the developed creep-type Se-60MSe cathode. Figure 6a shows the specific capacity of CT-ASS Se-60MSe cathode at 70°C at a rate of 0.1 C. To demonstrate the potential of all-electrochemical-active Se-MSe cathode without inactive solid electrolyte and conductive carbon, we estimate the energy density of the Se-60MSe cathode at the electrode level, which displays a volumetric energy density of 2460 Wh/L (Fig. 6b, detailed information for calculation shows in Table S7. Fig. S30). Moreover, due to the excellent kinetic of the Se-60MSe cathode, it can deliver a high capacity of 321.9 mAh/g for 5 mg/cm² (1.61 mAh/cm², Fig. S31) and 286 mAh/g for 10 mg/cm² (2.86 mAh/cm²), respectively. Even increased to 20 mg/cm², the capacity remains at 264 mAh/g (5.28 mAh/cm²). Meanwhile, the Se-60MSe cathode represents an excellent rate capability where the discharge capacity can reach 155 mAh/g at 2 C and recover to 229 mAh/g at a reduced rate of 0.2 C (Fig. 6c). Most importantly, our creep-type ASSB displays a remarkable long-term cycling life to survive more than 3000 cycles at 0.5 C. It is speculated that when the rate rapidly changes from 0.1 C to 0.5 C, the capacity decay in the initial cycles may originate from the contact failure of some biphasic reaction sites inside the electrode. Nonetheless, time-dependent Se creep slowly improves the interfacial contact and releases stress, resulting in nearly no capacity loss in the following long-term cycling (Fig. 6d). Such enhancement in cycling performance is further confirmed in the electrodes with higher mass loadings of 5 mg/cm² (Fig. S32) and 10 mg/cm² (Fig. S33). When the operation temperature is reduced from 70 °C to 25 °C, the CT-ASS Se-60MSe cathode still releases a capacity of 200 mAh/g (Fig. S34) with a stable long-cycle life of 1400 cycles (Fig. S35), indicating that the CT-ASS cathode has a wide range of operation temperature that covers most practical applications.

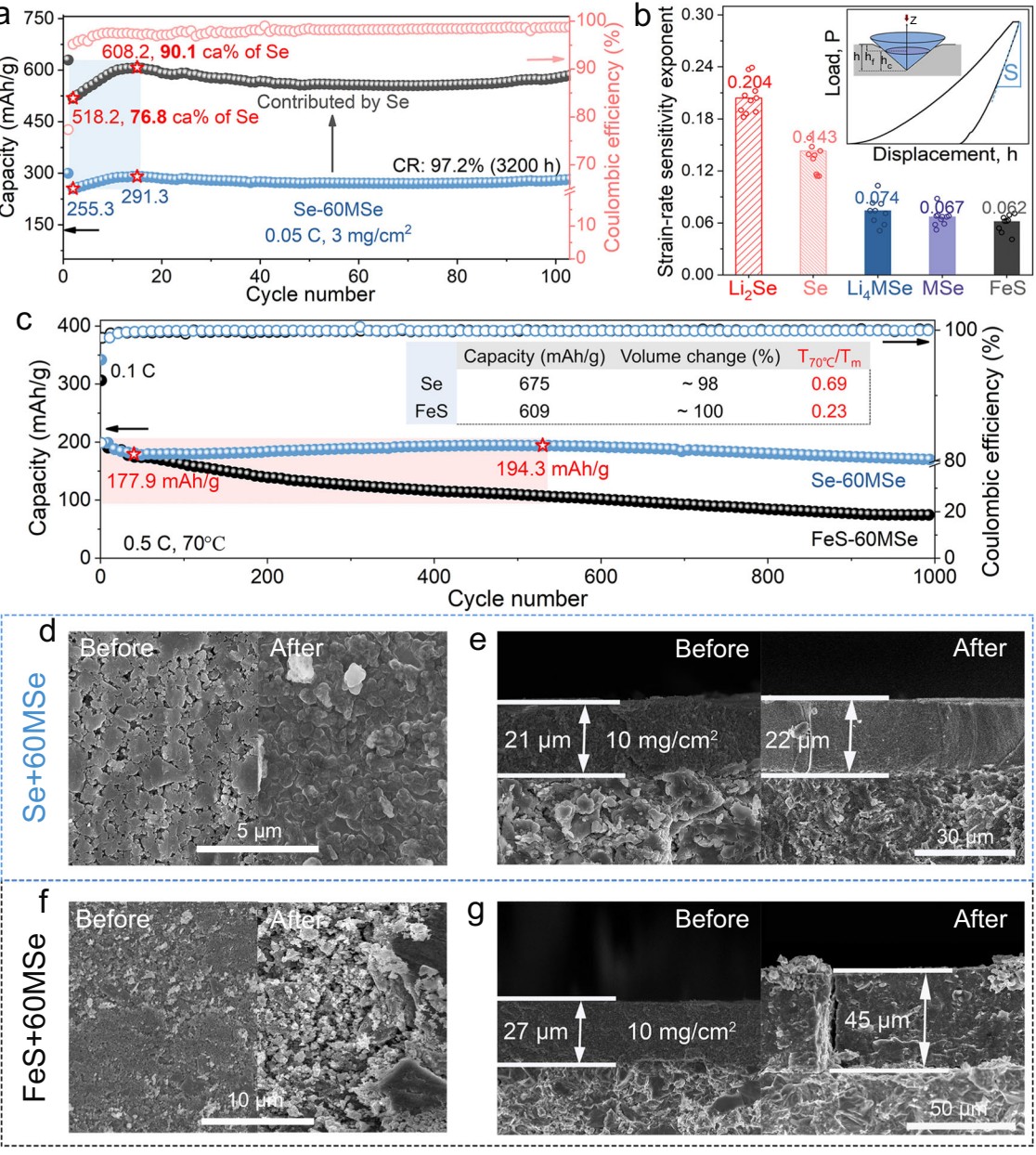

**Fig. 5 | Comparative investigation of the effect of creep on electrochemical performances. a** The cycling performance of CT-ASS Se-60MSe cathode at 0.05 C (1 C = 0.96 mA/cm²) and the corresponding capacity contributed by Se. **b** Comparison of the strain-rate sensitivity exponent, m. The inset shows the schematic principle of nanoindentation. **c** Comparison of cycling performance of Se-60MSe and FeS-60MSe cathodes. The inserted table shows the parameters comparison of Se and FeS. **d**, **f** The surface and **e**, **g** cross-sectional SEM images of Se-MSe and FeS-MSe cathodes are compared before and after 100 cycles (Fig. S29).

To evaluate the electrochemical performance of the Se-60MSe cathode in pouch cell, we assemble a single-layer pouch cell (Se-60MSe|Li₆PS₅Cl|InₓLi, 26.5 mAh, 4 cm × 4 cm, 1.66 mAh/cm²) using the wet-slurry-process for Se-60MSe cathode with 1 wt% binder. We adopt the conventional fixture for testing the commercial Li-ion batteries, with the applied pressure on the cell about 10 MPa (Fig. S36). This stack pressure is much lower than that used in the cold-pressing fixture for ASSBs testing at the lab scale. Nevertheless, our pouch cell can stably operate over 35 cycles (740 h) with negligible capacity fading. Benefiting from the high packing density of the cathode, the cell delivers a high gravimetric energy density of 658.3 Wh/kg and a volumetric energy density of 1749 Wh/L at the cathode level (Table S8). This reveals that our CT-ASS Se-60MSe cathode is compatible with mainstream manufacturing processes for Li-ion batteries, which

significantly reduces the cost of the technological transition from Li-ion batteries to ASSBs.

## Discussion

In conclusion, based on the specific chemomechanical characteristics of all-solid-state batteries, we have designed and developed innovative creep-type all-solid-state selenide alloy electrodes with a space-limited framework. In-situ SEM and numerical analysis are used to thoroughly investigate the (de)lithiation-stress-creep synergistic evolution and their effect on the electrochemical performance. We observed that the large volume strain of the Se from (de)lithiation is restricted by the rigid MSe framework, which generated stress gradients to induce the Se creep by diffusing into the pore space, accompanied by the internal stress released and the conformal interface formed after accumulation

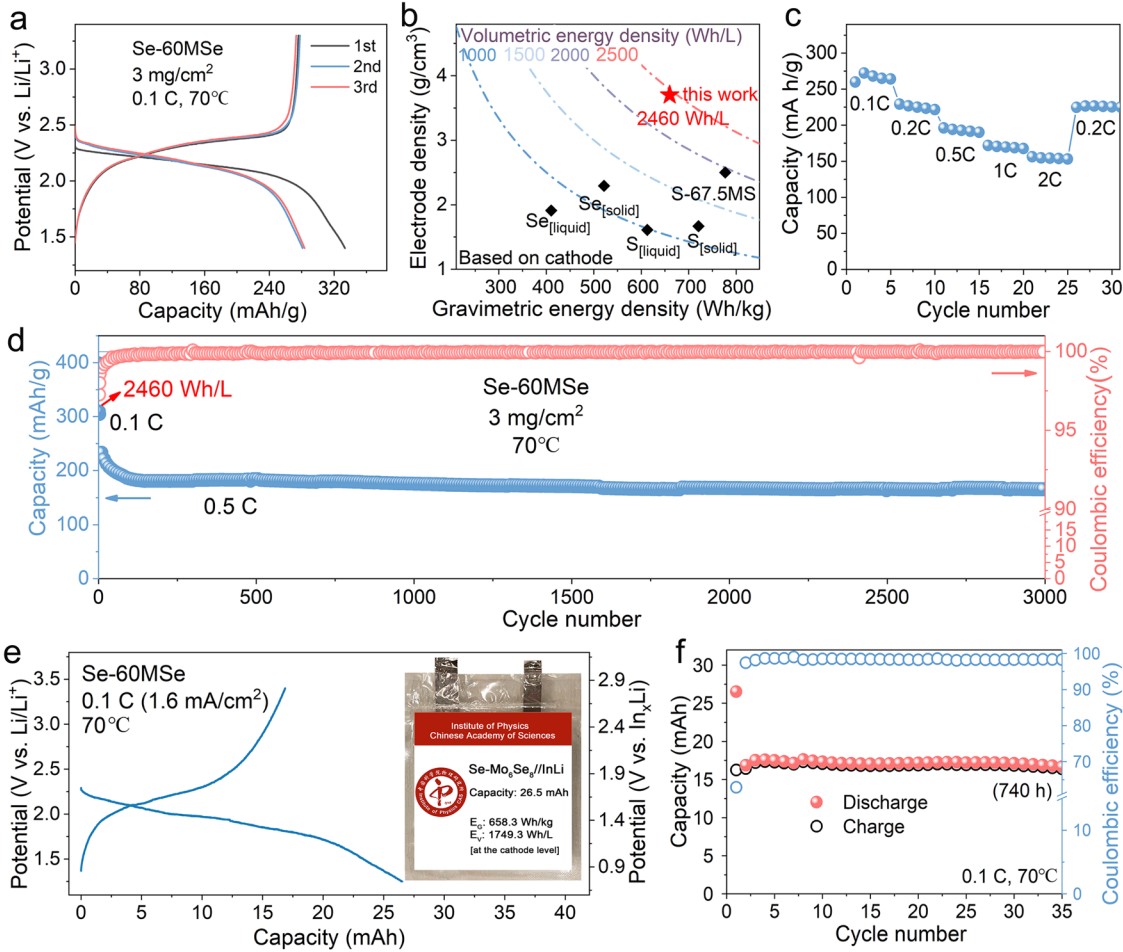

**Fig. 6 | Comprehensive electrochemical performance of creep-type Se-60MSe ASSBs. a** The galvanostatic curves for the Se-60MSe cathode. **b** The comparison of energy density between Se-60MSe cathode and other conventional Se[liquid][49], Se[solid][50], S[liquid][51], S[solid][52] and S-67.5MS[solid][53] cathodes. **c** The rate performance. **d** Long cycling stability of Se-60MSe cathode over 3000 cycles. **e** The charge-discharge curves and **f** the cycling performance of the developed Se-60MSe//InLi all-solid-state pouch cell. The inset shows the digital photo and the energy density of the pouch cell.

for repeated cycles. This optimizes the interparticle contact and forms a stable structure with stable long-term cycling performance. As a result, the creep-type Se-60MSe cathode displays a high volumetric energy density of 2460 Wh/L and a remarkable long-term cycling life for 3000 cycles at 0.5 C.

We believe that most high-capacity alloy electrode materials susceptible to creep at room temperature can pave the way for developing creep-type all-solid-state electrode systems. For other electrode materials with higher melting points, it is equally promising to develop creepable all-solid-state electrodes to achieve stable cycling performance by altering the electrode working temperature to obtain higher homologous temperature. In addition, as it is challenging to eliminate the stress inside all-solid-state electrodes, designing tailored-made electrode structures and cell configurations incorporating electrochemical-mechanical features of all-solid-state batteries may provide a new research avenue to achieve better electrochemical performance.

## Methods
### Materials
Se$_8$ (99.99%, Innochem), Li$_6$PS$_5$Cl (5-10 μm, Kejing star), Li (99.95%, 80 μm, CEL), FeS (99%, Alfa) and Indium (99.999%, 30 μm) were obtained commercially.

Chevrel phase Mo$_6$Se$_8$ was synthesized by leaching Cu from copper chevrel powder Cu$_2$Mo$_6$Se$_8$. First, MoSe$_2$ (99%, Innochem),

Cu (99%, Innochem), and Mo (99%, Innochem) powders in the molar ratio of 2:1:1 were ball-milled (QM-3SP04, NanDa Instrument) for 1 h, then the mixture along with iodine was pelleted and then sealed into Swagelok vessels, which were heated to 900 °C keeping for 48 h under Ar. Subsequently, the as-prepared Cu$_2$Mo$_6$Se$_8$ precursors were stirred in a 6 mol/L HCl solution for 12 h to extract Cu. Finally, the obtained powder (chevrel Mo$_6$Se$_8$) was washed with deionized water multiple times followed by drying at 120 °C overnight under vacuum.

Se-Mo$_6$Se$_8$ (Se-MSe) and FeS-Mo$_6$Se$_8$ (FeS-MSe) compounds were obtained by mixing Se or FeS with Mo$_6$Se$_8$ powders with different weight ratios by ball-milling at 300 rpm for 12 h.

Preparation of Li$_2$Se and Li$_4$Mo$_6$Se$_8$. Firstly, dissolve biphenyl (Bp) and lithium in dimethoxymethane (DME) with molar ratio of 1:1 and stir for 5 h to obtain a homogenous blue-black 1 mol/L Li-Bp/DME solution. Then add Li-Bp/DME solution to Se powders with the molar ratio of Li:Se = 2:1. After reacting under magnetic stirring for 7 h to ensure lithiation, the lithiated Se powders were collected by centrifugation, washing with DME, and vacuum drying. Add 1.6 mol/L n-butyllithium reagent in a molar ratio of Li:Mo = 2:3 to glass bottles containing Mo$_6$Se$_8$ powders. React under magnetic stirring for 7 h, the lithiated Mo$_6$Se$_8$ was collected by centrifugation, washing with n-hexane, and vacuum drying.

## Materials characterizations

The as-prepared samples were characterized by X-ray diffraction measurement using Cu Kα radiation (X' Pert Pro MPD) from 10° to 80° (2θ).

Scanning electron microscopy (SEM) images were recorded with a scanning electron microscope (Hitachi-s4800) under an accelerating voltage of 10 kV. In-situ SEM was employed to investigate the creep behavior of Se during the discharge-charge process. The cell assembly followed the standard procedure in a glove box, which was immediately transferred to a vacuum-protected in-situ SEM stage, which was then affixed to an SEM holder and linked to a potentiostat (LAND CT2001A). To avoid damage from the electron beam to the sample, SEM images were captured at 20-minute intervals during galvanostatic discharging-charging at a rate of 0.12 C.

The electronic conductivities of the samples were measured with the four-probe method under pressure from 9.55 MPa to 57.30 MPa at room temperature by Powder impedance measurement system (MCP PD51, Mitsubishi Chemical).

Porosity and density measurement of electrodes under different stress are manual tests. First, the height of the mold without powder was measured by an inside micrometer. Then, the cathode of a certain mass was poured into the casing mold and sequentially compressed with different pressure (100 MPa, 270 MPa, 360 MPa) and lasted 3 mins at every pressure. The height after every pressure was measured by a micrometer and the subtraction was the height of the cathode pellet. During cycling, the stack pressure applied on the cell which comes from the torque force between the screws and the nuts was measured by a stress sensor.

## Mechanics characterizations

The Se, $Li_2Se$, $Li_4Mo_6Se_8$ powders of 300-600 mg were pressed into a die (10 mm diameter) under an applied uniaxial compressive stress of 10 t (~1250 MPa) for 10 min to get the dense pellets of millimeter-scale thickness. Moreover, the $Mo_6Se_8$ and FeS powders of 300-600 mg were pressed into a die (10 mm diameter) and hot-pressed at 700 °C under 50 MPa for 30 mins. The thickness of the obtained samples was in the range of 1.5-3 mm, thick enough to obtain reliable mechanical data. To obtain polished samples for mechanical characterization, the solid samples were fixed into epoxy resin by cold inlay and attached to a hand-operated polishing tool. Next, the sample was polished using silicon carbide sandpaper with progressively smaller grit sizes (120, 500, 800, 1200, 2400, and 4000) and diamond polishing pads (3, 1, and 0.5 μm). Xylene, extra dry with molecular sieves (water ≤ 50 ppm, Innochem), was used to clean the samples and polishing tool after each polishing step because it was inactive in the samples. The entire sample is processed in a glove box ($H_2O$ < 0.1 ppm, $O_2$ < 0.1 ppm). To prevent sample exposure to air, the polished samples were placed into the specialized fluid cell, transferred from the glovebox to the indenter, and carefully mounted in the instrument to maintain full sample immersion.

Mechanical properties of Young's modulus (E), hardness (H) and Strain rate sensitivity exponent (m) were performed at the Nano Indenter G200 XP (Keysight) with a diamond Berkovich indenter. The displacement control mode was selected to measure E and H by maintaining a constant strain rate of 0.05 $s^{-1}$ superimposed on a continuous stiffness measurement (CSM) mode with an oscillation of 2 nm at 45 Hz. The maximum indentation depth was set at 2000 nm for sample $Mo_6Se_8$, 3000 nm for Se and FeS, and 4000 nm for $Li_2Se$ and $Li_4Mo_6Se_8$. The test indentations for every sample were at 10 distinct locations and center-to-center spacing of the indentations was 70 μm, more than 10 times the indentation depth (2-4 μm), sufficient to obtain accurate results for a Berkovich indenter[28].

From the load–depth hysteresis, E and H were calculated using the Oliver-Pharr method[29].

$$H = \frac{P_{max}}{A_{(h_c)}} \tag{1}$$

$$E_r = \frac{S\sqrt{\pi}}{2\beta\sqrt{A}} \tag{2}$$

$$\frac{1}{E_r} = \frac{1-\upsilon^2}{E} + \frac{1-\upsilon_i^2}{E_i} \tag{3}$$

Where $P_{max}$ was directly obtained from the maximum load and A was the calibration area function of the indenter in contact depth $h_c$. $E_r$ was defined as the equivalent elastic modulus. S was the slope of the unloading curve and β was a known dimensionless constant that depends on the geometry of the indenter (β of Berkovich =1.034). E was converted from the calculation of $E_r$, which considered the deformation of both the indenter and sample. υ was Poisson's ratio of material. The elastic modulus and Poisson's ratio ($E_i$, $\upsilon_i$) of the Berkovich indenter was 1141 GPa and 0.07. Poisson's ratio of samples was assumed to be 0.25.

Strain rate sensitivity test. The nanoindentation testing technique called "strain-rate jump tests" was utilized to measure the strain-rate sensitivity of materials, where a standard CSM method was adapted to perform several abrupt changes in the applied strain rate at defined indentation depths during one single indentation[11,30–32]. For performing the strain-rate jumps on the samples, the indentation strain rate was kept constant up to an indentation depth of ∼2000 nm. Afterward, changes in the strain rate were applied every 300 nm. Four different strain rates (from 0.12 to 0.004 $s^{-1}$) were used during a single indentation experiment with a 5 nm amplitude and 45 Hz oscillation.

The steady-state creep strain rate which can be empirically related with rupture time by Monkman-Grant equation[33] is known to be strongly dependent on the applied stress σ, (absolute) temperature T, and grain size d:

$$\dot{\varepsilon} = f\left(\frac{b}{d}\right)^p \left(\frac{\sigma}{G}\right)^n \exp\left(-\frac{Q}{RT}\right) \tag{4}$$

where f is a material and temperature-related factor, G is the shear modulus, b is the magnitude of Burgers vector, Q is the activation energy for creep, R is the gas constant, p is the inverse grain-size exponent, and n is the creep stress exponent. In this equation, the stress exponent, $n = (\partial\ln\dot{\varepsilon})/(\partial\ln\sigma)$, is often considered as a valuable indicator for the predominant creep mechanism. For each indentation test in this Berkovich indenter indentation, the mean stress σ and the strain rate $\dot{\varepsilon}$ are often considered as[34,35],

$$\sigma \propto H = \frac{P_{max}}{\psi h^2} \tag{5}$$

$$\dot{\varepsilon} = \frac{1}{h}\frac{dh}{dt} = \frac{1}{2}\left(\frac{\dot{P}}{P} - \frac{\dot{H}}{H}\right) \approx \frac{1}{2}\frac{\dot{P}}{P} \tag{6}$$

Here Ψ is the constant related with tip geometry (e.g., 24.56 for the Berkovich tip). The strain rate sensitivity exponent (m) at constant temperature can be described as,

$$m_{nanoindentation} = \frac{d\ln H}{d\ln\dot{\varepsilon}_{nanoindentaion}} \tag{7}$$

Then the strain rate sensitivity exponent m is obtained from the slope of that ln (H)/ln (ε̇) curve. The creep stress exponent n can be obtained by n = 1/m.

The constant load tests. Tests followed a typical chronology: Indenter loads at a constant loading rate until the given peak load, holds at the peak load (100 mN and 200 mN) for an extended dwell time (100 s), then partially unloads to 10% of the peak load and holds for a second dwell time (60 s) to measure the drift-rate, and finally completely withdraws from the sample.

## Electrochemical measurements

The ASSBs are assembled in an argon-filled glovebox. First, 100 mg $Li_6PS_5Cl$ solid electrolyte (SE) powder was transferred into a cell casing mold with an inner diameter of 10 mm and was uniaxially compressed with pressure about 270 MPa for 3 mins. The Se-MSe powder was applied onto one side of the SE layer as cathode followed by a uniaxial pressure of 360 MPa for 3 mins. Then, $In_{1.3}Li$ alloy anode (0.6 V vs Li/Li⁺)[36] was obtained by cold-pressing the In foil (99.99%, Φ 10 mm, thickness 100 μm) and Li foil (99.99%, Φ 9 mm, thickness 80 μm) together under the pressure of 100 MPa for 3 mins. Finally, the $In_{1.3}Li$ alloy was attached to the other side of the SE pellet, followed by a uniaxial pressure of 360 MPa for 3 mins to get the Se-MSe|LPSC|InLi ASSBs. The initial stack stress applied on the battery by the torque force from the screws is about 100 MPa measured by the stress sensor shown in Fig. S1. The mass loadings of the cathode were from 3 mg/cm² to 20 mg/cm². The electrochemical performances were estimated on the LAND battery test station (LAND CT2001A). All tests are at 70 °C unless otherwise specified. Cyclic voltammograms (CVs) were carried out with a CHI660E electrochemical workstation (CH Instruments Ins.) at 25 °C.

The diffusion coefficients of Se-MSe samples with different mixing proportions were obtained from the galvanostatic intermittent titration technique (GITT). These samples were discharged/charged at a rate of 0.05 C for 1 h and relaxed at an open circuit for 2 h to equilibrium potentials, testing from 0.9 V to 3.2 V (vs. InxLi).

Fabrication of all-solid-state pouch cell: comprise 99 wt.% Se-60MSe active materials and 1 wt.% PVDF binder in N-Methyl pyrrolidone by wet-mixing slurry, and then coat the as-prepared slurry on current-collector of Al foils with mass loading of ~5 mg/cm², followed by drying under vacuum at 120 °C to attain the cathode. To fabricate the $Li_6PS_5Cl$ membrane, 99 wt% $Li_6PS_5Cl$, and 1 wt% PTFE were weighed and thoroughly mixed in an agate mortar and then roll-pressed into a membrane with a thickness of ~150 μm. Then, the Se-60MSe cathode layer (4 cm × 4 cm), $Li_6PS_5Cl$ membrane (5 cm × 5 cm), and $In_{1.24}Li$ anode (0.6 V vs Li/Li⁺, In foil, 4 cm × 4 cm, thickness 30 μm; Li foil, 4 cm × 4 cm, thickness 20 μm) were pressed together to form an all-solid-state pouch cell. The initial stack pressure of the fixture to the pouch cell is about 10 MPa.

## Numerical simulation

To simulate the mechanical behaviors of composite electrodes, we build a two-dimensional plane-strain model consisting of two components—active material and electrolyte material, where lithiation only occurs in the active material. The symmetric boundary conditions were applied to the surrounding boundaries of the 2D model to constrain the free deformation of the composite cathode (Fig. 4a). Before the battery cycling, both components are in a stress-free state. We ignore the (de)lithiation heterogeneity in the electrode as we use a relatively slow C-rate of 0.1 C[37,38].

Therefore, the lithiation state and volume change of active particles are assumed to be proportional to the (de)lithiation time. The deformation and stress in the electrode are caused by the mechanical interactions between the active material and electrolyte material. The governing equation for the mechanical equilibrium is represented as $\nabla \cdot \mathbf{S} = 0$, where $\mathbf{S}$ is the nominal stress calculated as $\mathbf{S} = \mathbf{K}{:}\boldsymbol{\varepsilon_e}$, $\mathbf{K}$ the stiffness matrix, and $\boldsymbol{\varepsilon_e}$ the elastic strain. The total deformation gradient $\mathbf{F}$ of a representative material element is considered as $\mathbf{F} = \mathbf{F_e} \cdot \mathbf{F_l} \cdot \mathbf{F_{cr}}$, where $\mathbf{F_e}$ represents the reversible elastic deformation, $\mathbf{F_l}$ the lithiation-induced volumetric deformation, and $\mathbf{F_{cr}}$ the irreversible creep

deformation. The lithiation-induced volumetric change is $1 + \Omega C$, where $\Omega$ and C are the partial molar volume and concentration of Li, respectively. The constitutive behavior of the electrolyte material is described by a linear elastic material model, while the active materials are assumed to the linear elastic materials with time-dependent creep behavior, as described by the Norton model[39],

$$\dot{\varepsilon}_{cr} = A \left( \frac{\sigma_e}{\sigma_{ref}} \right)^n \tag{8}$$

where the $\dot{\varepsilon}_{cr}$ is the creep strain rate, A the creep rate coefficient, n the stress exponent, $\sigma_{ref}$ a reference stress level and $\sigma_e$ the equivalent stress (i.e., von Mises Stress). A representative volume element (RVE) is reconstructed in which the volume fractions of Se and MSe are adjusted to be consistent with the experiment (Se: 35.6 vol.%, MSe: 39.7 vol.%, porosity: 24.7 vol.%). All the material properties used in the numerical simulation are summarized in Table S6.

## Data availability

All data generated or analyzed during this study are provided in this paper (and its supplementary files). Source data are provided in this paper.

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

## Acknowledgements

All authors acknowledge the support of the CAS Youth Interdisciplinary Team and the Center for Clean Energy. R.X. acknowledges the support of the National Natural Science Foundation of China (12302232).

## Author contributions

L.S. conducted the project, and conceived the concept and the manuscript's writing. X.X. contributed to the conceptualization of the study, experiments including the electrochemical experiments, in-situ SEM and mechanical tests, data analysis, data interpretation, and the manuscript's writing. R.X conducted the numerical simulation, data analysis, and the manuscript's writing. T.L. assisted in the materials' structural characterization. C.T. contributed to the pre-lithiation of the material. G.J. participated in the manufacture of pouchcell. H.L. and L.C. contributed to the revision of the manuscript. All authors participated in the manuscript reviewing and editing.

## Competing interests

The authors declare no competing interests.
