## [Peer Review File · Nature Communications]

REVIEWER COMMENTS

Reviewer #1 (Remarks to the Author):

In this manuscript, Suo et al presented a strategy to deal with the significant chemomechanical issue by exploiting material creep under the lithiation-induced stress to achieve stable cycling performance. The author fabricated the Se-Mo₆Se₈ cathode as a prototype, in which the material Se could creep into the pore to improve the interparticle contact and dissipative stress with the mixing ionic/electronic conductor Mo₆Se₈ as the rigid stress self-constraining framework. The authors provide an extensive analysis in experimental data and characterizations to support their claim, including as mechanical testing, in-situ SEM investigation of morphology, numerical simulations, and the comprehensive assessment of electrochemical performance. The presented electrochemical performances look promising, and this strategy appears to be applicable to a wide range of high-capacity creepable alloy electrodes. In addition, this study may motivate further bottom-up research toward developing targeted materials, electrode structures, and cell structures based on the mechanical and structural nature of all-solid-state batteries. I would like to recommend its acceptance by Nature Communications after minor revisions to the following comments.

1. The authors assembled 40wt% Se-60wt% Se-Mo₆Se₈ cathode to demonstrate that the creep mechanism favors better cycling performance. According to Fig. S8, the 40Se-60MSe and 50Se-50MSe mixing ratios have similar discharge capacity and the 50Se-50MSe has a higher theoretical capacity, why was the 40-60 ratio ultimately selected?
2. According to the Fig. 2c, the Li diffusion coefficient of Li_xMo₆Se₈ is lower than oxide electrolyte LLZO and sulfide electrolyte LGPS, the unsatisfactory Li diffusion coefficient seems to be contradictory to the well-remained capacity of the Se-MSe cathode at a rate of 2C (Fig. 6c) and the high mass loadings of 20 mg/cm² (Fig. S22).
3. The numerical simulations (Fig. 4) were performed based on the parameters of unlithiated Se and MSe. Given that Se will alloy during lithiation, it is more appropriate to reflect the creep behavior by integrating the parameters of lithiated Li₂Se.
4. In p.6 l.142 “we would expect the pore volume to re-expand to nearly 78% of its original volume.” How is the calculation for '78%' done with the CE value?
5. Why the label of electrolyte layer in figure 1C is LGPS? However, the subsequent experiments by the author used LPSC as the electrolyte.

Reviewer #2 (Remarks to the Author):

The authors have provided an interesting in-situ virtualization of the volume change during charge/discharge process of the conversion type of MoS-based cathode in ASSBs. However, given some incorrect technical statements (especially from mechanics aspects) and critical questions that have to be answered, I do not think this manuscript is ready for a publication yet at this moment. Given the novelty and experimental designs with the simulation inputs, I believe this manuscript can be resubmitted and much suitable for other journal(s). My comments are as follows:

1. As the authors in the abstract stating this is promising for a practical application, however, a stack pressure of 100 MPa during battery cycling is impossible for a commercialized cell. It

is required a stack pressure of at most 10 MPa. Have the authors observed obvious creep behavior under ~ 10 MPa?

2. Given the fact that authors were using MoS-based conversion-type materials, the huge volume change would occur during multiple cyclings. Have said that, the authors were actually using a torque force to maintain an "external" cycling pressure of 100 MPa, this led to severe drop of stack pressure and thus led to an inaccuracy of stack pressure control during cell cycling test, especially for a long-term cycling. That may mean that, the creep behavior was only happening in the few first cycles rather than the entire cycling. The authors have to make sure the stack pressure maintained precisely consistent to lead to a conclusion this improved performance was due to the creep behavior instead of some other chemical/electrochemical modification.

3. From commercialization aspect, obtaining energy density from cathode is a tricky. To make it a practical indication, the authors at least need to list the mass and volume contribution from all the components inside the cells (cathode, LPSCI, anode etc. optional: current collector etc.) And then replace the Figure 6b with the suggested approach and put Figure 6b somewhere properly in the supporting information.

4. As the authors indicated in Figure 6, for the elevated temperature tests, the authors have to be extremely careful that the elevated temperature (70 C) would cause the chemical decomposition of LPSCI electrolyte, although the decomposition kinetics is not relatively fast. But the issue is that the authors have tested a sufficiently long time, for example (Figure 6f, 740 hours), that led to a situation that the generated decomposed product at the electrode/LPSCI and the decomposition of LPSCI would led to a huge volume change. Therefore, only if the authors can defensively stated that it was the creep behavior of the cathode that led to the improved performance with excluding these critical factors.

5. There is little information found for the nano-indentation experimental method. There is no loading/unloading profiles (with unit and values, loading forces, loading rate and displacement) and also lack of nano-indentation data for the fused quartz sample. This led to lots of difficulty. For example, at this stage, it is impossible for reviewers to evaluate if the 50um spacing is high enough. And how thick is the samples for nano-indentation? In addition, was the nano-indentation performed in the glovebox or somewhere else? All these things affect results.

6. In Figure S18, the authors stated the "Modulus", that "Modulus" referred to what "Modulus"?

7. In Table S6, it is not clear to me how the authors got Young's modulus information, for example, the 19.5 GPa for Se, given the fact of the huge porosity that the authors have, authors must have a clear explanation why the authors can get such high Young's modulus for Se and MSe. The authors did not mention the critical factor-particle sizes of the materials used for nano-indentation.

8. In the videos attached, the authors claimed that this volume change was due to the creep behavior and SEM is a localized characterization (not sure how large the overall volume change would be), both of these factors would contribute to a factor that this behavior could actually due to the volume change of the conversion cathode nature. As the authors simply replaced the Se or MSe with sulfur cathode, even though you do not apply a stack pressure, these similar phenomena can also be seen. This led to a thought/assumption that this

visualization was simply due to the volume change because of the electrochemical redox and/or was due to contributions from both electrochemical redox and creep that was not sure/clear if electrochemical redox or creep contributed more.

9. Having said all the above, the authors need to emphasize what is new in this manuscript as lots of publications have used Se/MSe or similar electrode to prepare for an ASSBs in a similar system, because from mechanics perspective, it is not clear to me this is a mechanics-driven design and the mechanics impact is also not clear in the design.

Reviewer #3 (Remarks to the Author):

The authors design the cathode for lithium cells utilizing creep/stress relaxation properties of Se to reduce the stress and avoid fracture in active material. While the idea is sound, the implementation raises some questions.

1. While the elemental pure Se has high homologous temperature, alloying with lithium changes that. Li₂Se has much higher melting temperature than Se. Since the melting point of lithium selenide is ~ 1300 degC it is very surprising that the authors found extremely low yield strength of this material (very low hardness).
2. Nanoindentation curves show waviness that needs explanation. This waviness is more pronounced in materials with higher modulus and hardness (i.e. FeS vs Se). There could be something wrong with the instrument that produces these periodic oscillations. The hardness of lithium selenide seems close to zero meaning the material deforms plastically almost immediately. The authors should supply the SEM images of residual hardness impressions from nanoindentation experiments of all materials involved. Were the nanoindentation tests done in a glovebox?
3. It is unclear why the authors emphasize closure of the pores in Se cathode as “striking observation”. Upon lithiation there is a ~ 98% volume expansion of Se so the reduction of pores should not be a surprise.
4. What is the “time-dependent hysteresis” that the authors mention on line 189?
5. In supplementary materials the nano indentation figures are captioned as “Resulting hardness(modulus) in the nanoindentation strain-rate jump experiment.” What is the strain-rate jump experiment? The description of nano indentation experiments (albeit brief) states that they were done in CSM mode. Could the authors elaborate how the “strain rate jumps” were incorporated into the experiments and what was their purpose?
6. Boundary conditions of the numerical simulations should be clearly explained (especially considering that a 3D problem was replaced with a 2D one).
7. The manuscript requires substantial re-write to improve the language.

Based on the above, a major revision is recommended before the manuscript can be considered for any further review.

Response to Reviewer #1

Comment: In this manuscript, Suo et al. presented a strategy to deal with the significant chemomechanical issue by exploiting material creep under the lithiation-induced stress to achieve stable cycling performance. The author fabricated the Se-Mo₆Se₈ cathode as a prototype, in which the material Se could creep into the pore to improve the interparticle contact and dissipative stress with the mixing ionic/electronic conductor Mo₆Se₈ as the rigid stress self-constraining framework. The authors provide an extensive analysis in experimental data and characterizations to support their claim, including as mechanical testing, in-situ SEM investigation of morphology, numerical simulations, and the comprehensive assessment of electrochemical performance. The presented electrochemical performances look promising, and this strategy appears to be applicable to a wide range of high-capacity creepable alloy electrodes. In addition, this study may motivate further bottom-up research toward developing targeted materials, electrode structures, and cell structures based on the mechanical and structural nature of all-solid-state batteries. I would like to recommend its acceptance by Nature Communications after minor revisions to the following comments.

Reply: We highly appreciate the reviewer's positive comments. All your concerns are point-to-point addressed.

Q1. The authors assembled 40wt% Se-60wt% Se-Mo₆Se₈ cathode to demonstrate that the creep mechanism favors better cycling performance. According to Fig. S8, the 40Se-60MSe and 50Se-50MSe mixing ratios have similar discharge capacity and the 50Se-50MSe has a higher theoretical capacity, why was the 40-60 ratio ultimately selected?

A1. We thank the reviewer for the comments. As shown in Fig. S8, although the theoretical volumetric energy density of 50Se-50MSe cathode is slightly higher than that of 40Se-60MSe with similar discharge capacity, the *practical volumetric energy density* of 40Se-60MSe is higher than that of 50Se-50MSe due to the higher utilization of Se (indicated by the solid-circle curve in Fig. S8 (b)). This higher utilization of Se indicates that Se is more thoroughly utilized, making it more susceptible to Se creep to improve interparticle contact and dissipate stress with better cycling performance. In this circumstance, the ratio of Se and MSe should present an optimized value, i.e., 40/60, in this work.

Supplementary Materials:

Fig. S8 | Comparison of electrochemical performance of Se-MSe with different mixing ratios. (a) The galvanostatic charge-discharge curves of Se-MSe composite electrodes for the 2nd cycle. (b) Corresponding volumetric energy density and active capacity utilization of Se for Se-MSe electrodes. Active capacity utilization of Se is calculated as shown in part Calculation of Energy density.

Q2. According to the Fig. 2c, the Li diffusion coefficient of $\text{Li}_x\text{Mo}_6\text{Se}_8$ is lower than oxide electrolyte LLZO and sulfide electrolyte LGPS; the unsatisfactory Li diffusion coefficient seems to be contradictory to the well-remained capacity of the Se-MSe cathode at a rate of 2C (Fig. 6c) and the high mass loadings of 20 mg/cm² (Fig. S22).

A2. We thank the reviewer for the comments. In our constructed two-phase Se-MSe cathode, electrochemical reactions can occur all around the two-phase interface instead of triple-phase heteropoints in the traditional all-solid-state cathode with active materials, carbon (electronic conductor) and solid electrolyte (ionic conductor). Therefore, the effective reaction sites can be significantly increased, which improves the cycling performance at a high rate and high mass loading. In addition, the electrons and ions share the same transport channel inside the mixing ionic/electronic Mo_6Se_8 for a two-phase Se-MSe cathode, which dramatically reduces the tortuosity compared to that of a three-phase cathode in which separate electron/ion conductors force carriers to bypass each other's conductors with increased transport path. The detailed mechanism has been demonstrated in our previous work in Ref. 37 in Supplementary Materials (ACS Energy Lett. 7, 766–772, 2022). Combining these merits, although the Li diffusion coefficient of $\text{Li}_x\text{Mo}_6\text{Se}_8$ is slightly lower than that of solid electrolytes such as LLZO and

LGPS, our proposed Se-MSe cathode exhibits faster kinetics and thus retains good electrochemical performance even at a high rate of 2C (Fig. 6c) and the high mass loading of 20 mg/cm² (Fig. S27).

Supplementary Materials: “-----There are more electrochemical reaction sites in the two-phase cathode and where ions and electrons migrating in the same medium in Se-MSe cathodes significantly reduce the tortuosity factor, benefiting the cathode kinetics with an excellent performance even at high mass loadings and great current densities. -----”

Q3. The numerical simulations (Fig. 4) were performed based on the parameters of un lithiated Se and MSe. Given that Se will alloy during lithiation, it is more appropriate to reflect the creep behavior by integrating the parameters of lithiated Li₂Se.

A3. We thank the reviewer for the comments. We have reformed the numerical simulation using the parameters of Se and lithiated Li₂Se. The general “rule of mixture” is used to determine the evolution of mechanical properties of the Li_xSe during lithiation and delithiation. For example. The Youngs’ Module of Li_xSe is expressed by,

$$E_{\text{Li}_x\text{Se}} = C E_{\text{Li}_2\text{Se}} + (1 - C) E_{\text{Se}}$$

Where *C* is the normalized lithiated state ranging from 0 (i.e., Se) to 1 (i.e., Li₂Se), other parameters, such as the creep stress component of the Li_xSe, are determined by a similar approach.

The updated Fig.4 is presented as follows,

Fig. 4. Numerical analysis of the creep behavior of the all-solid-state cathodes. (a) The RVE for the Se-MSe cathode. (b) The deformation and equivalent stress

in the fully lithiated Se-MSe cathode. (c) The creep strain's magnitude (color contour) and direction (arrows) in the Se particles. The particle contact ratio (d) and average equivalent stress (e) in the Se-MSe and LCO-MSe cathodes upon galvanostatic cycling. (f) The equivalent stress and creep strain in the CT-ASS Se-MSe cathode at different cycles.

The following discussions have been updated in the revised manuscript:

“In contrast, the Se-Se and Se-MSe contact ratios in the Se-MSe cathode can reach 70.5% and 67.0% in the first cycle and gradually increase to 84.4% and 79.5% in the 20th cycle, which adds effective electrochemical site and increases the continual electron and ion paths inside the cathode. From the structural integrity perspective, the average equivalent stress developed in the Se and MSe particles inside the Se-MSe cathode is only 0.099 GPa and 0.171 GPa, respectively, and further drops to 0.067 GPa and 0.156 GPa at the 20th cycle.”

As our instinctive expectation, the Li₂Se presents the mechanical properties between the Se and Li, based on the general “rule of mixtures.” This means that the Li₂Se tends to deform and creep more than the Se because Li is very soft and creepable. By comparing the simulation results (as shown by the following Fig. R1) from the models with and without integrating the parameters of lithiated Li₂Se, we indeed notice that the particle-particle contact is further improved and the stress in the cathode is further reduced if the parameters of lithiated Li₂Se is integrated. It well demonstrates the viability of our strategy to utilize the creep to boost the stability of the all-solid-state cathode.

Fig. R1 | Comparison of the simulation results from the models with and without integrating the parameters of lithiated Li₂Se.

Q4. In p.6 l.142 “we would expect the pore volume to re-expand to nearly 78% of its original volume.” How is the calculation for '78%' done with the CE value?

A4. We thank the reviewer for the comments. The coulombic efficiency (CE) is obtained by discharge (lithiation) capacity divided by charge (delithiation) capacity. That is, if plastic deformation is not considered, after one cycle of charging and discharging, the degree of lithiation of Se is 22% compared to the

initial state, i.e., the volume of Se increases by about 22% ($\Delta V_{\text{Se/Li}_2\text{Se}}=98\%$), approximating the relative shrinkage of the pores by 22%. This implies that at the end of charging, the pore size is approximately 78% of the initial state (Fig. S10). However, most of the actual pore sizes after the end of the charge are much smaller than 78% of the initial state (Fig. 3d in main text), indicating that Se creeps into the pores to induce the pores to shrink further. Thus, the volume change of porosity can correspond to the volume change, even though there will be minor variances in the local environment, and we present the change of 20 pores to emphasize static tendency.

Main text: “-----As shown in Fig. 3c-II, III, the pores undergo a distinct shrinkage with an evident enhancement in interparticle contact during discharge (lithiation), and the Se particles retain their structural integrity without being pulverized or broken. During subsequent charge (delithiation), Li-ion is extracted from Se particles, theoretically inducing Se retraction and pore enlargement. Supposedly, if there is no irreversible creep deformation, the pores volume should re-expand to 78% of the initial volume (Fig. S10), because the degree of lithiation of Se remains at 22% (calculated by the Columbic efficiency (CE)) at the end of charge (Fig. 3c-V). However, the actual porosity further contracts during charge (delithiation, Fig. 3c-IV, V), implying that the particles continue to deform into the pores due to Se creep. This phenomenon can be exclusively ascribed to the creep deformation induced by the internal stresses exerted on the Se particles by the rigid Mo_6Se_8 framework. We calculate the relative areas of 20 individual pores to substantiate our observations further. The results reveal that most pore sizes are much smaller than 78% of the initial state, and eight pores undergo further shrinkage during delithiation (Fig. 3d, Fig. S11), highlighting the pervasive nature of Se/ Li_2Se creep within the electrode structure. This is a direct observation that Se creeps throughout the charging and discharging process, forming improved contact and leasing stress in the electrode.-----”

Supplementary Materials:

Fig. S10 | Schematic of the volume change of Se particles during (de)lithiation and the corresponding pore size.

“The coulombic efficiency (CE) is obtained by discharge (lithiation) capacity divided by charge (delithiation) capacity. That is, if plastic deformation is not considered, after one cycle of charging and discharging, the degree of lithiation of Se is 22% compared to the initial state, i.e., the volume of Se increases by about 22% ($\Delta V_{\text{Se/Li}_2\text{Se}}=98\%$), approximating the relative shrinkage of the pores by 22%. This is to imply that at the end of charging, the pore size is approximated to be 78% of the initial state. Thus, the volume change of porosity can be used to correspond to the volume change to validate the creep mechanism, even though there will be minor variances in the local environment, and we present the change of 20 pores to emphasize static tendency.”

“-----If without plastic deformation, the electrode pores theoretically shrink during discharge (lithiation) and re-expand to 78% of the volume of the initial state after end of charge (delithiation). However, most of the actual pore sizes after the end of charge are much smaller than 78% of the initial state (Fig. 3d in main text), indicating that Se creep into the pores to induce the pores further shrink.”

Q5. Why the label of electrolyte layer in figure 1C is LGPS? However, the subsequent experiments by the author used LPSC as the electrolyte.

A5. Thank the reviewer very much for pointing out that. We have corrected the schematic Fig.2c from LGPS to LPSC.

Fig. 2. Construct CT-ASS cathodes. (a) The phase diagram of deformation mechanisms for materials at different stress states and temperatures²⁵⁻³⁰. Particles inside ASSBs are subjected to shear stress greater than the external stack pressure (Fig. S1)⁵. (b) Comparison of the performance metrics of common cathode materials^{19,31}. (c) The structure diagram of the creep-type Se-60MSe cathode. The inset shows the ionic and electronic conductivity of MSe^{32,33}. (d) Comparison of the porosity and density of Se-50SE-10SP and Se-60MSe all-solid-state cathodes.

Response to Reviewer #2

Comment: The authors have provided an interesting in-situ virtualization of the volume change during charge/discharge process of the conversion type of MoS-based cathode in ASSBs. However, given some incorrect technical statements (especially from mechanics aspects) and critical questions that have to be answered, I do not think this manuscript is ready for a publication yet at this moment. Given the novelty and experimental designs with the simulation inputs, I believe this manuscript can be resubmitted and much suitable for other journal(s). My comments are as follows.

Reply: We highly appreciate the reviewer's positive comments and constructive suggestions that are helpful for us to improve the quality of our manuscript further. All your concerns are point-to-point addressed. We hope the revised manuscript can be suitable for publication on *Nature Communications*.

Q1. As the authors in the abstract stating this is promising for a practical application, however, a stack pressure of 100 MPa during battery cycling is impossible for a commercialized cell. It is required a stack pressure of at most 10 MPa. Have the authors observed obvious creep behavior under ~ 10 MPa?

A1. We thank the reviewer for the comments. The creep deformation rate depends on the material mechanical parameters (yield strength, shear modulus, etc.), homologous temperature, working time, and applied load (Ref. 9 in main text, *The Journal of Chemical Physics* 111, 1999). Even at room temperature, the homologous temperature of Se is rather high, making it highly susceptible to creep. Therefore, we believe the Se creep persists even at the stack pressure of 10 MPa. However, compared to other plastic deformation like fracture, creep strain is more difficult to track directly given the slight and time-dependent strain rate that we designed most of our experiments under the high stack pressure of 100 MPa and at 70 °C precisely to amplify the effect of Se creep on electrochemical properties and electrode structure. Two pieces of evidence can support our claim:

1) We performed the cycling test of the pouch cell at the stack pressure of about 10 MPa (Fig. 6d). It presents a capacity increase at the first 200 h and excellent cycling stability up to 740 h, primarily due to the creep of Se particles can indirectly prove our claim.

2) Moreover, we performed an operando stack pressure test on the pouch cell with an initial stack pressure of 9.76 MPa (Fig. S33), and there is also a capacity increase during the first 20 cycles (Fig. S33) similar to in Fig. 6d. Therefore, we believe that the Se creep is still available even at the stack pressure of 10 MPa. Note that the internal stress, subject to the combination of external stack pressures and stress fluctuations generated by volume strain of materials during

(de)lithiation, is the key factor determining the creep of Se particles, instead of external stack pressure.

Supplementary materials:

Fig. S33 | Operando stress measurement for Se-60MSe|LPSC|InLi (3cm×3cm, 1.48 mAh/cm²) all-solid-state pouch cell. (a) Galvanostatic curves of the battery. (b) Galvanostatic cycling of the battery along with the measured stack-pressure changes. The pressure at t = 0 is 9.76 MPa.

Q2. Given the fact that authors were using MoS-based conversion-type materials, the huge volume change would occur during multiple cycling. Have said that, the authors were actually using a torque force to maintain an "external" cycling pressure of 100 MPa, this led to severe drop of stack pressure and thus led to an inaccuracy of stack pressure control during cell cycling test, especially for a long-term cycling. That may mean that, the creep behavior was only happening in the few first cycles rather than the entire cycling. The authors have to make sure the stack pressure maintained precisely consistent to lead to a conclusion this improved performance was due to the creep behavior instead of some other chemical/electrochemical modification.

A2. We thank the reviewer for the comments. We agree the reviewer that the stack pressure applied by the screws might fluctuate during cycling. Therefore, we supplemented the operando pressure test on the battery (Fig. S16) to track the evolution of stack pressure upon battery cycling. The stack pressure presents a 4.5 MPa drop during the first cycle (22 h) due to initial mechanical relaxation, while

the stack pressure maintains above 90 MPa throughout the cycles. This minor stress drop has little effect on total stack pressure and is high enough to constrain the expansion of the Se particles and thus incur the creep of Se for contact improvement and stress relaxation.

Moreover, even though the tack pressure drops (ΔP still within 10 MPa) for the long-term cycling of about 1000 h, the electrode capacity continues to increase in this period (Fig. 5a, 5c). Combined with the characterizations that the electrode did not undergo other chemical/electrochemical side reactions (please see our answer to Question #4), we believe the creep of Se particles can occur for a relatively long time instead of in the first several cycles.

Supplementary materials:

Fig. S16 | Operando pressure measurement for Se-60MSe|LPSC|InLi ASSB. (a) Galvanostatic curves of the battery. (b) Galvanostatic cycling of the battery along with the measured stack-pressure changes. The pressure at $t=0$ is 101.2 MPa.

Q3. From commercialization aspect, obtaining energy density from cathode is a tricky. To make it a practical indication, the authors at least need to list the mass and volume contribution from all the components inside the cells (cathode, LPSCl, anode etc. optional: current collector etc.) And then replace the Figure 6b with the suggested approach and put Figure 6b somewhere properly in the supporting information.

A3. We thank the reviewer for the comments. We concur with the reviewer that comparing the energy density of the electrodes is rarely offers much information

for estimating the energy density of a practical battery. Currently, most all-solid-state batteries (ASSBs) used in research are investigated in steel shell devices or pouch cell to demonstrate the electrochemical performance of electrode materials or electrolytes at the lab scale, where the energy density of the ASSBs is hardly calculated due to the challenge of manufacturing an electrolyte film that is sufficiently thin (<50 μm) and highly performing. Thus, We primarily aim to demonstrate the potential of all-electrochemical-active Se-MSe cathode for future applications: on the one hand, due to the absence of low-density inactive solid electrolyte and conductive carbon, the high-density and high-capacity Se-60MSe cathode could display a high volumetric energy density; on the other hand, the lack of solid electrolyte enables all-solid-state cathode manufactured by the commercialized wet-slurry-process in air, as shown in Fig. 6e, 6f for the Se-60MSe|Li₆PS₅Cl|In_xLi pouch cell. Combining these merits, we are confident that the energy density of the Se-MSe cathode by a mature manufacturing process can further surpass that of our hand-prepared pouch cell in future practical applications. To clarify to the reader that we are only trying to elaborate on the potential of Se-MSe cathode, we have made the following corrections in the main text.

Main text: “----- To demonstrate the potential of all-electrochemical-active Se-MSe cathode without inactive solid electrolyte and conductive carbon, we estimate the energy density of the Se-60MSe cathode at the electrode level, which displays a volumetric energy density of 2460 Wh/L (Fig.6b, detailed information for calculation shows in Table S7. Fig.S27). -----”

“To evaluate the electrochemical performance of the Se-60MSe cathode in pouch cell-----”

Q4. As the authors indicated in Figure 6, for the elevated temperature tests, the authors have to be extremely careful that the elevated temperature (70 C) would cause the chemical decomposition of LPSCl electrolyte, although the decomposition kinetics is not relatively fast. But the issue is that the authors have tested a sufficiently long time, for example (Figure 6f, 740 hours), that led to a situation that the generated decomposed product at the electrode/LPSCl and the decomposition of LPSCl would led to a huge volume change. Therefore, only if the authors can defensively state that it was the creep behavior of the cathode that led to the improved performance with excluding these critical factors.

A4. We thank the reviewer for the comments. We present the Coulombic efficiency of Se-60MSe|LPSC|InLi ASSBs cycled at 70 °C for 7800 h (Fig. R2), which is close to 100% and has almost no decay with cycling, illustrating that LPSC decomposition is almost negligible when cycled at 70°C. Additionally, according to the comparison of differential capacity dQ/dV curves in Fig. R3, no characteristic peak of the decomposition voltage of LPSC was observed in the operating voltage range of the Se-60MSe|LPSC|InLi battery, which also indicates

that there is no decomposition of LPSC. Furthermore, we assembled a control battery steel|LPSC|InLi|steel tested at 70°C for 200 cycles, where the steel|LPSC interface is equated to the original Se-MSe|LPSC interface. And it is visible that the battery has almost no available capacity (Fig. R4).

Fig. R2 | The Coulombic efficiency of Se-60MSe|LPSC|InLi ASSBs over 3000 cycles (7800 h) at 70°C.

Fig. R3 | Comparison of the differential capacity dQ/dV curve of (a) the Se-60MSe|LPSC|InLi battery and (b) the oxidation and first reduction curve of LPSC from Ref. (Nature Materials, 19, 428-435, 2020). [Schwietert, T.K., Arszewlewska, V.A., Wang, C. et al. Clarifying the relationship between redox activity and electrochemical stability in solid electrolytes. *Nat. Mater.* 19, 428–435 (2020), reproduced with permission from SNCSC.]

Fig. R4 | Galvanostatic cycling of the control battery steel|LPSC|InLi|steel at 70°C.

Q5. There is little information found for the nano-indentation experimental method. There is no loading/unloading profiles (with unit and values, loading forces, loading rate and displacement) and also lack of nano-indentation data for the fused quartz sample. This led to lots of difficulty. For example, at this stage, it is impossible for reviewers to evaluate if the 50um spacing is high enough. And how thick is the samples for nano-indentation? In addition, was the nano-indentation performed in the glovebox or somewhere else? All these things affect results.

A5. We thank the reviewer very much for the comments. Due to limited experiment conditions without nanoindentation apparatus in the glove box, the prior nanoindentation was performed in air (Fig. R5). To evaluate the mechanical properties more rigorously, referred to Ref. 38 (Adv. Energy Mater., 7, 1602011, 2017), *we have retested all samples using a fluid cell that protected the samples in the mineral oil (Fig. S19).* The mechanical properties for the samples tested in the mineral oil are consistent with the those tested in the air (Fig. S21-S25). Meanwhile, we have added detailed information about sample's preparation and nanoindentation test including test patterns, test parameters, loading/unloading profiles as well as data analysis methods into the **Supplementary Materials Mechanics Characterizations** part.

For the nanoindentation in air, we examined beforehand the samples' stability in air to demonstrate the data reliability, and there was no substantial change in their structure by XRD patterns after 20 hours (Fig. R5). Therefore, we consider the nanoindentation tests in air are reliable since the test time for each sample was less than 3 hours. However, to be more rigorous, we have updated all the results of mechanical tests by the new data in the revised manuscript (Fig. 5b, Fig. S18-S25).

Fig. R5 | Samples' stability in air for 20 h.

P. Sudharshan Phani and W.C. Oliver had verified that a minimum indent spacing of 10 times the indentation depth is sufficient to obtain accurate results for a Berkovich indenter (Ref. 2 in Supplementary Materials, Materials & Design 164, 107563, 2019). Therefore, an indentation spacing of 50 μm was selected to gather reliable data with the indentation depth between 2-4 μm in our testing. To avoid controversy, the indentation spacing was altered to 70 μm for the retest, and the results of retest resembled to the previous test. All samples prepared for nanoindentation test have a thickness of 1-3 mm (Fig. R6). For indentation depths at micrometer level, the sample thickness is large enough to avoid the substrate effect during nanoindentation test.

Fig. R6 | Thickness of samples for nanoindentation.

The detailed revisions and data are as follows:

Main text:

Fig. 5. (b) Comparison of the strain-rate sensitivity exponent, m . The inset shows the schematic principle of nanoindentation.

Supplementary Materials, Mechanics Characterizations:

The Se, Li₂Se, Li₄Mo₆Se₈ powders of 300-600 mg were pressed into a die (10 mm diameter) under an applied uniaxial compressive stress of 10 t (~1250 MPa) for 10 min to get the dense pellets of millimeter-scale thickness. Moreover, the Mo₆Se₈ and FeS powders of 300-600 mg were pressed into a die (10 mm diameter) and hot-pressed at 700 °C under 50 MPa for 30 min. The thickness of the obtained samples was in the range of 1.5-3 mm, thick enough to obtain reliable mechanical data. To obtain polished samples for mechanical characterization, the solid samples were fixed into epoxy resin by cold inlay and attached to a hand-operated polishing tool. Next, the sample was polished using silicon carbide sandpaper with progressively smaller grit sizes (120, 500, 800, 1200, 2400, and 4000) and diamond polishing pads (3, 1, and 0.5 μm). Xylene, extra dry with molecular sieves (water ≤ 50 ppm, Innochem), was used to clean the samples and polishing tool after each polishing step because it was inactive in the samples. The entire sample is processed in a glove box (H₂O < 0.1 ppm, O₂ < 0.1 ppm). To prevent sample exposure to air, the polished samples were placed into the specialized fluid cells, transferred from the glovebox to the indenter, and carefully mounted in the instrument to maintain full sample immersion.

Mechanical properties of Young's modulus (E), hardness (H), and Strain rate sensitivity exponent (m) were performed at the Nano Indenter G200 XP (Keysight) with a diamond Berkovich indenter. The displacement control mode was selected to measure E and H by maintaining a constant strain rate of 0.05 s^{-1} superimposed on a continuous stiffness measurement (CSM) mode with an oscillation of 2 nm at 45 Hz. The maximum indentation depth was set at 2000 nm for sample Mo₆Se₈, 3000 nm for Se and FeS, and 4000 nm for Li₂Se and Li₄Mo₆Se₈. The test indentations for every sample were at 10 distinct locations, and center-to-center spacing of the indentations was 70 μm, more than 10 times the indentation depth (2-4 μm), sufficient to obtain accurate results for a Berkovich indenter (Ref. 2 in Supplementary Materials, Materials & Design 164, 107563, 2019).

From the load–depth hysteresis, E and H were calculated using the Oliver-Pharr method³.

$$H = \frac{P_{max}}{A(h_c)}$$

$$E_r = \frac{S\sqrt{\pi}}{2\beta\sqrt{A}}$$

$$\frac{1}{E_r} = \frac{1 - \nu^2}{E} + \frac{1 - \nu_i^2}{E_i}$$

Where P_{max} was directly obtained from the maximum load and A was the calibration area function of the indenter in contact depth h_c . E_r was defined as the equivalent elastic modulus. S was the slope of the unloading curve and β was a known dimensionless constant that depends on the geometry of the indenter (β of Berkovich = 1.034). E was converted from the calculation of E_r , which considered the deformation of both the indenter and sample. ν was Poisson's ratio of material. The elastic modulus and Poisson's ratio (E_i, ν_i) of the Berkovich indenter was 1141 GPa and 0.07. Poisson's ratio of samples was assumed to be 0.25.

Strain rate sensitivity test. A nanoindentation testing technique called "strain-rate jump tests" was utilized to measure the strain-rate sensitivity of materials, where a standard CSM method was adapted to perform several abrupt changes in the applied strain rate at defined indentation depths during one single indentation.⁴⁻⁷ For performing the strain-rate jumps on the samples, the indentation strain rate was kept constant up to an indentation depth of ~ 2000 nm. Afterward, changes in the strain rate were applied every 300 nm. Four different strain rates (from 0.12 to 0.004 s⁻¹) were used during a single indentation experiment with a 5 nm amplitude and 45 Hz oscillation.

The steady-state creep strain rate which can be empirically related with rupture time by Monkman-Grant equation (Ref. 8 in Supplementary Materials, Mechanical metallurgy, 3, 1976) is known to be strongly dependent on the applied stress σ , (absolute) temperature T , and grain size d :

$$\dot{\epsilon} = f \left(\frac{b}{d}\right)^p \left(\frac{\sigma}{G}\right)^n \exp\left(-\frac{Q}{RT}\right)$$

Where f is a material- and temperature-related factor, G is the shear modulus, b is the magnitude of Burgers vector, Q is the activation energy for creep, R is the gas constant, p is the inverse grain-size exponent, and n is the creep stress exponent. In this equation, the stress exponent, $n = (\partial \ln \dot{\epsilon}') / (\partial \ln \sigma)$, is often considered as a valuable indicator for the predominant creep mechanism. For each indentation test in this Berkovich indenter indentation, the mean stress σ and the strain rate $\dot{\epsilon}$ are often considered as,

$$\sigma \propto H = \frac{P_{max}}{\Psi h^2}$$

$$\dot{\epsilon} = \frac{1}{h} \frac{dh}{dt} = \frac{1}{2} \left(\frac{\dot{P}}{P} - \frac{\dot{H}}{H} \right) \approx \frac{1}{2} \frac{\dot{P}}{P}$$

Here Ψ is the constant related with tip geometry (e.g., 24.56 for the Berkovich

tip). And the strain rate sensitivity exponent (m , simply by $m=1/n$) at constant temperature can be described as (Ref. 9 in Supplementary Materials, Acta Materialia 54, 7, 2006; Ref. 10 in Supplementary Materials, Journal of Materials Research 19, 51, 2004),

$$m_{\text{nanoindentation}} = \frac{d \ln H}{d \ln \dot{\epsilon}_{\text{nanoindentation}}}$$

Then the strain rate sensitivity exponent m is obtained from the slope of that $\ln(H)/\ln(\dot{\epsilon})$ curve.

Fig. S18 | Digital photo of polished samples for nanoindentation. Black zone is sample, and transparent part is epoxy resin for cold inlay with sample fixed.

Fig. S19 | Schematic of specialized fluid cell with mineral oil to prevent the sulfide sample from being exposed to air during nanoindentation.

Fig. S20 | Nanoindentation strain-rate jump experiment by four different applied strain rates at corresponding displacement.

Fig. S21 | Mechanical characteristics of Se sample by nanoindentation test. (a-c) Displacement control mode for measurement of E and H by maintaining a constant strain rate of 0.05 s^{-1} . Using the Oliver-Pharr method, (a) the corresponding load–displacement curves and the calculated (b) E and (c) H. (d-f) Nanoindentation strain-rate jump experiment with four different applied strain rates. (d) Corresponding load–displacement curves, (e) the measured H, and (f) the calculated strain-rate sensitivity exponent m for Se. (g, h) Residual topography after nanoindentation experiments by microscopy. Scale bar: 100 μm for (g), 10 μm for (h).

Fig. S22 | Mechanical characteristics of Li_2Se sample by nanoindentation test. (a-c) Displacement control mode for measurement of E and H by maintaining a constant strain rate of 0.05 s^{-1} . Using the Oliver-Pharr method, (a) the corresponding load–displacement curves and the calculated (b) E and (c) H . (d-f) Nanoindentation strain-rate jump experiment with four different applied strain rates. (d) Corresponding load–displacement curves, (e) the measured H , and (f) the calculated strain-rate sensitivity exponent m for Li_2Se . (g, h) Residual topography after nanoindentation experiments by microscopy. Scale bar: 50 μm for (g), 10 μm for (h).

Fig. S23 | Mechanical characteristics of MSe sample by nanoindentation test. (a-c) Displacement control mode for measurement of E and H by maintaining a constant strain rate of 0.05 s^{-1} . Using the Oliver-Pharr method, (a) the corresponding load–displacement curves and the calculated (b) E and (c) H. (d-f) Nanoindentation strain-rate jump experiment with four different applied strain rates. (d) Corresponding load–displacement curves, (e) the measured H, and (f) the calculated strain-rate sensitivity exponent m for MSe. (g, h) Residual topography after nanoindentation experiments by microscopy. Scale bar: 100 μm for (g), 10 μm for (h).

Fig. S24 | Mechanical characteristics of Li_4MSe sample by nanoindentation test. (a-c) Displacement control mode for measurement of E and H by maintaining a constant strain rate of 0.05 s^{-1} . Using the Oliver-Pharr method, (a) the corresponding load–displacement curves and the calculated (b) E and (c) H . (d-f) Nanoindentation strain-rate jump experiment with four different applied strain rates. (d) Corresponding load–displacement curves, (e) the measured H , and (f) the calculated strain-rate sensitivity exponent m for Li_4MSe . (g, h) Residual topography after nanoindentation experiments by microscopy. Scale bar: 100 μm for (g), 10 μm for (h).

Fig. S25 | Mechanical characteristics of FeS sample by nanoindentation test. (a-c) Displacement control mode for measurement of E and H by maintaining a constant strain rate of 0.05 s^{-1} . Using the Oliver-Pharr method, (a) the corresponding load–displacement curves and the calculated (b) E and (c) H. (d-f) Nanoindentation strain-rate jump experiment with four different applied strain rates. (d) Corresponding load–displacement curves, (e) the measured H, and (f) the calculated strain-rate sensitivity exponent m for FeS. (g, h) Residual topography after nanoindentation experiments by microscopy. Scale bar: 100 μm for (g), 10 μm for (h).

Q6. In Figure S18, the authors stated the "Modulus", that "Modulus" referred to what "Modulus"?

A6. We thank the reviewer very much for pointing out that. We have corrected it to Young's modulus in Fig.S21-S25 for the retested experimental results.

Q7. In Table S6, it is not clear to me how the authors got Young's modulus information, for example, the 19.5 GPa for Se, given the fact of the huge porosity that the authors have, authors must have a clear explanation why the authors can get such high Young's modulus for Se and MSe. The authors did not mention the critical factor-particle sizes of the materials used for nano-indentation.

A7. We thank the reviewer for pointing out this omission. The Young's modulus of Se (18 GPa for retest) and MSe (37.6 GPa for retest) in Table S6 is obtained from nanoindentation tests in Fig. S20 and Fig. S22, whereas the Young's modulus of LCO (191 GPa) in Table S6 is derived from Ref. 35 (Journal of Asian Ceramic

Societies, 5, 2, 2017). Here, The Se, Li_2Se , $\text{Li}_4\text{Mo}_6\text{Se}_8$ samples for nanoindentation are cold-pressed by an applied uniaxial compressive stress of 10 t (~1250 MPa) for 10 min, and the Mo_6Se_8 and FeS samples are hot-pressed at 700 °C under 50 MPa for 30 min. The samples obtained in this way are similar to the bulk solid ones due to the very high density and low porosity (450 mg of Se powder was cold pressed into a 10 mm diameter sample with a thickness of 1.246 mm and a density of 4.60 mg/cm², i.e., porosity of 4.35%), thus the measured Young's modulus and hardness are close to the intrinsic Young's modulus and hardness of the material. Models constructed for simulation are performed by taking Young's modulus and hardness in Table S6 as intrinsic ones by adjusting porosity consistent with the experimental electrode (Se: 35.6 vol.%, MSe: 39.7 vol.%, porosity: 24.7 vol.%). The numerical analysis in Fig. 4 and Fig. S12-15 has been revised based on the retested nanoindentation data. The particle size of MSe (~2 μm), Se (~2 μm), and FeS (~1 μm) are respectively displayed in Fig. S4 and Fig. S26, and SEM images of Li_2Se (5-10 μm) and Li_4MSe (3-5 μm) were added to Fig. S17b, c. There is no visible size effect for these particles during nanoindentation.

Supplementary Materials: “The Young's modulus of Se, Li_2Se , MSe, and Li_4MSe listed in Table S6 are obtained from the nanoindentation test in Fig. S21-S24, and the Young's modulus of LCO is from the previous reports³⁵.”

Fig. S17 | Structural and morphological characterization for Li_2Se and Li_4MSe obtained by chemical prelithiation. (a) XRD patterns of Li_2Se and Li_4MSe . SEM images of (b) Li_2Se and (c) Li_4MSe .

Q8. In the videos attached, the authors claimed that this volume change was due to the creep behavior and SEM is a localized characterization (not sure how large the overall volume change would be), both factors would contribute to a factor that this behavior could actually be due to the volume change of the conversion cathode nature. As the authors simply replaced the Se or MSe with sulfur cathode, even though you do not apply a stack pressure, these similar phenomena can also be

seen. This led to a thought/assumption that this visualization was simply due to the volume change because of the electrochemical redox and/or was due to contributions from both electrochemical redox and creep that was not sure/clear if electrochemical redox or creep contributed more.

A8. We thank the reviewer for the comments. To investigate the microscopic morphology and contact evolution of particles during charging and discharging, the electron microscope was magnified 15k times with an observation area of approximately 8 μ m*6 μ m shown in the video. The Se-60MSe cathode only contains of Se and MSe, with a theoretical 98% volume change for Se during alloying with Li and an 11.2% volume change for MSe during Li-ion insertion/extraction (Table S2). If no Se creep, SEM may show that the particles expand due to lithiation with pores shrinking during discharge and particles contract due to delithiation with re-enlarged pores during charging. By contrast, the electrode pores actually appear to further contract during charging (delithiation), according to the *in-situ* SEM observations (Fig. 3c-IV, V). This implies that the particles continue to deform into the pores even during charging due to Se/Li₂Se creep. Thus, we hypothesize that the pore contraction during discharge arises from particle expansion by electrochemical redox and Se/Li₂Se creep, whereas the pore further contraction during charging is attributed to Se/Li₂Se creep. That is, Se creeps throughout the charging and discharging process with improved contact and relaxed stress.

Main text: “----- As shown in Fig. 3c-II, III, the pores undergo a distinct shrinkage with an evident enhancement in interparticle contact during discharge (lithiation), and the Se particles retain their structural integrity without being pulverized or broken. During subsequent charge (delithiation), Li-ion is extracted from Se particles, theoretically inducing Se retraction and pore enlargement. Supposedly, if there is no irreversible creep deformation, the pores volume should re-expand to 78% of the initial volume (Fig. S10), because the degree of lithiation of Se remains at 22% (calculated by the Columbic efficiency (CE)) at the end of charge (Fig. 3c-V). However, the actual porosity further contracts during charge (delithiation, Fig. 3c-IV, V), implying that the particles continue to deform into the pores due to Se creep. This phenomenon can be exclusively ascribed to the creep deformation induced by the internal stresses exerted on the Se particles by the rigid Mo₆Se₈ framework. We calculate the relative areas of 20 individual pores to substantiate our observations further. The results reveal that most pore sizes are much smaller than 78% of the initial state, and eight pores undergo further shrinkage during delithiation (Fig. 3d, Fig. S11), highlighting the pervasive nature of Se/Li₂Se creep within the electrode structure. This is a direct observation that Se creeps throughout the charging and discharging process, forming improved contact and leasing stress in the electrode.”

Q9. Having said all the above, the authors need to emphasize what is new in this manuscript as lots of publications have used Se/MSe or similar electrode to prepare for an ASSBs in a similar system, because from mechanics perspective, it is not clear to me this is a mechanics-driven design and the mechanics impact is also not clear in the design.

A9. We thank the reviewer very much for the comments. Though Se/MSe has been studied in solid-state batteries before, the creepable characteristics of these alloys and how these mechanical characteristics affect electrochemical performance have received little attention. To date, the single-phase Li/Na metal anode has been subjected to creep engineering to prevent stress concentration. But because there isn't a creep occurrence environment in the cathode's multi-phase porous architecture, the creep-type all-solid-state cathode hasn't been observed. In this work, we first propose to exploit the lithiation-induced stress multi-phase porous cathode to drive the time-dependent creep of active materials (AM). We have designed and developed innovative creep-type all-solid-state selenide alloy electrodes with an incompressible stress self-constraining framework, by which the mismatch strains between AM and solid electrolyte (SE) can be accommodated by the creep strain, leading to the dynamic reshaping of the conformal AM/SE interfaces and the continual relaxation of mismatch stresses in the electrodes. Our proposed creep-type all-solid-state cathode concept could serve as a meaningful guide to future mechanics-driven design for solid-state battery research, especially screening electrode materials with distinct mechanical features and targeting structural design to achieve the mechanical regulation on electrochemical performance. Additionally, our developed *in-situ* SEM observation, operando stack pressure test as well as numerical analysis of the creep behavior are all extremely essential for the research of the electrochemical-mechanical coupling behavior inside ASSBs. We strongly believe this new strategy in electrode design that actively utilizes, instead of passively adapting, the mechanical stresses in electrodes to boost their electrochemical performance can address the complex electrochemical-mechanical degradations in ASSBs, which paves the way to realizing ASSBs for practical applications. Meanwhile, our pioneer exploration of stabilizing the structural integrity of ASSBs by manipulating the interplays between electrochemistry and mechanics provides valuable new insights toward the future design of next-generation ASSBs with optimum performance.

Response to Reviewer #3

Comment: The authors design the cathode for lithium cells utilizing creep/stress relaxation properties of Se to reduce the stress and avoid fracture in active material. While the idea is sound, the implementation raises some questions.

Reply: We highly appreciate the reviewer's positive comments and constructive suggestions that are helpful for us to improve the quality of our manuscript further. All your concerns are point-to-point addressed.

Q1. *While the elemental pure Se has high homologous temperature, alloying with lithium changes that. Li₂Se has much higher melting temperature than Se. Since the melting point of lithium selenide is ~ 1300 degC it is very surprising that the authors found extremely low yield strength of this material (very low hardness).*

A1. We thank the reviewer very much for the comments. To be honest, we are also in confusion about this discrepancy. We informed from the Ref. (Solid State Ionics, 325, 1, 2018) that Li₂Se has a high melting point of about 1300°C (Fig. R7). However, results of nanoindentation test (either test in air or supplemented test with mineral oil protection, Fig. S21, 22) demonstrate that Li₂Se (E, 12.7 GPa; H, 0.32 GPa; m, 0.204) has a lower modulus and hardness than Se (E, 18 GPa; H, 0.96 GPa; m, 0.143), as well as a bigger strain-rate sensitivity exponent (i.e., tend to creep). As our instinctive expectation, the Li₂Se should present the mechanical properties between the Se and Li, based on the general “rule of mixtures”. This means that the mechanical properties of Li₂Se measured by the nanoindentation seem to be reasonable, as Li is very soft and creepable.

During the manuscript revision, we also tested the thermal stability of Li₂Se by thermogravimetric analysis and differential scanning calorimetry (TG-DSC, 900°C is the instrumental temperature limit), as shown in Fig. R8. The melting point of Li₂Se is indeed higher than 900 °C, but there are two unknown heat absorption peaks at 110 °C and 238 °C. The crystal structure and morphology of Li₂Se were investigated after heating at 120°C and 250°C for 30 mins (Fig. R9). The treatment at 250°C did improve the particle's morphology and crystallinity while did not alter the crystal structure of Li₂Se. We believe that there may be some interesting and unknown phenomena that incur the discrepancy about the thermal stability of Li₂Se.

Fig. R7 | Melting temperatures T_m of compounds Li_2X ($\text{X} = \text{chalcogen}$) from Ref. *Solid State Ionics*, 325, 1, 2018. [Reprint from *Solid State Ionics*, Volume 325, Julius Schneider, Thorsten Schröder, Markus Hoelzel, Oliver Kluge, Wolfgang W. Schmahl, Oliver Oeckler, Phase transitions to superionic Li_2Te and Li_2Se – A high-temperature neutron powder diffraction study, atom displacements, probability density functions and atom potentials, p 90-101, Copyright (2018), with permission from Elsevier.]

Fig. S21 | Mechanical characteristics of Se sample by nanoindentation test. (a-c) Displacement control mode for measurement of E and H by maintaining a constant strain rate of 0.05 s^{-1} . Using the Oliver-Pharr method, (a) the corresponding load–displacement curves and the calculated (b) E and (c) H . (d-f) Nanoindentation strain-rate jump experiment with four different applied strain rates. (d) Corresponding load–displacement curves, (e) the measured H , and (f) the

calculated strain-rate sensitivity exponent m for Se. (g, h) Residual topography after nanoindentation experiments by microscopy. Scale bar: 100 μm for (g), 10 μm for (h).

Fig. S22 | Mechanical characteristics of Li_2Se sample by nanoindentation test. (a-c) Displacement control mode for measurement of E and H by maintaining a constant strain rate of 0.05 s^{-1} . Using the Oliver-Pharr method, (a) the corresponding load–displacement curves and the calculated (b) E and (c) H . (d-f) Nanoindentation strain-rate jump experiment with four different applied strain rates. (d) Corresponding load–displacement curves, (e) the measured H , and (f) the calculated strain-rate sensitivity exponent m for Li_2Se . (g, h) Residual topography after nanoindentation experiments by microscopy. Scale bar: 50 μm for (g), 10 μm for (h).

Fig. R8 | Thermal stability of (a) Li_2Se and (b) Se by TG-DSC.

Fig. R9 | The crystal structure and morphology change of Li_2Se after heat treating for 30 mins at 120°C and 250°C.

Q2. Nanoindentation curves show waviness that needs explanation. This waviness is more pronounced in materials with higher modulus and hardness (i.e. FeS vs Se). There could be something wrong with the instrument that produces these periodic oscillations. The hardness of lithium selenide seems close to zero meaning the material deforms plastically almost immediately. The authors should supply the SEM images of residual hardness impressions from nanoindentation experiments of all materials involved. Were the nanoindentation tests done in a glovebox?

A2. We thank the reviewer very much for the comments. This waviness of nanoindentation curves is due to that we perform the “strain-rate jumps” approach to perform several abrupt changes in the applied strain rate at defined indentation depths during one single indentation. The “strain rate jumps” testing technique was developed by Verena Maier to measure the strain-rate sensitivity of materials (Ref. 7 in supplementary materials, J. Mater. Res., 26, 11, 2011, *Nanoindentation strain-rate jump tests for determining the local strain-rate sensitivity in nanocrystalline Ni and ultrafine-grained Al*), where a standard CSM method was adapted to perform several abrupt changes in the applied strain rate at defined indentation depths during one single indentation.

Due to limited experiment conditions without nanoindentation apparatus in the glove box, the prior nanoindentation was performed in air (Fig. R5). To evaluate the mechanical properties more rigorously, referred to Ref. 38 (Adv. Energy Mater., 7, 1602011, 2017), *we have retested all samples using a fluid cell that protected the samples in the mineral oil (Fig. S19)*. The mechanical properties for the samples tested in the mineral oil are consistent with the those tested in the air (Fig. S21-S25). Meanwhile, to better clarify the details of the “strain-rate jumps” test, we have added information about sample preparation and nanoindentation test including test patterns, test parameters, loading/unloading profiles as well as data analysis methods into the **Supplementary Materials Mechanics Characterizations** part.

For the nanoindentation in air, we examined beforehand the samples' stability in air to demonstrate the data reliability, and there was no substantial change in their structure by XRD patterns after 20 hours (Fig. R5). Therefore, we consider the nanoindentation tests in air are reliable since the test time for each sample was less than 3 hours. However, to be more rigorous, we have updated all the results of mechanical tests by the new data in the revised manuscript (Fig. 5b, Fig. S18-S25).

Fig. R5 | Samples' stability in air for 20 h.

As shown in the Fig. S21 (e), the hardness of lithium selenide reaches a steady state value of about 0.5 GPa. The high-resolution optical images of the residual nanoindentation impressions of all materials involved have been added into the Supplementary Materials. We can clearly observe the triangle impressions induced by the indenter.

The detailed revisions and data are as follows:

Main text:

Fig. 5. (b) Comparison of the strain-rate sensitivity exponent, m . The inset shows the schematic principle of nanoindentation.

Supplementary Materials, Mechanics Characterizations:

The Se, Li₂Se, Li₄Mo₆Se₈ powders of 300-600 mg were pressed into a die (10 mm diameter) under an applied uniaxial compressive stress of 10 t (~1250 MPa) for 10 min to get the dense pellets of millimeter-scale thickness. Moreover, the Mo₆Se₈ and FeS powders of 300-600 mg were pressed into a die (10 mm diameter) and hot-pressed at 700 °C under 50 MPa for 30 min. The thickness of the obtained samples was in the range of 1.5-3 mm, thick enough to obtain reliable mechanical data. To obtain polished samples for mechanical characterization, the solid samples were fixed into epoxy resin by cold inlay and attached to a hand-operated polishing tool. Next, the sample was polished using silicon carbide sandpaper with progressively smaller grit sizes (120, 500, 800, 1200, 2400, and 4000) and diamond polishing pads (3, 1, and 0.5 μm). Xylene, extra dry with molecular sieves (water ≤ 50 ppm, Innochem), was used to clean the samples and polishing tool after each polishing step because it was inactive in the samples. The entire sample is processed in a glove box (H₂O < 0.1 ppm, O₂ < 0.1 ppm). To prevent sample exposure to air, the polished samples were placed into the specialized fluid cells, transferred from the glovebox to the indenter, and carefully mounted in the instrument to maintain full sample immersion.

Mechanical properties of Young's modulus (E), hardness (H), and Strain rate sensitivity exponent (m) were performed at the Nano Indenter G200 XP (Keysight) with a diamond Berkovich indenter. The displacement control mode was selected to measure E and H by maintaining a constant strain rate of 0.05 s⁻¹ superimposed on a continuous stiffness measurement (CSM) mode with an oscillation of 2 nm at 45 Hz. The maximum indentation depth was set at 2000 nm for sample Mo₆Se₈, 3000 nm for Se and FeS, and 4000 nm for Li₂Se and Li₄Mo₆Se₈. The test indentations for every sample were at 10 distinct locations, and center-to-center spacing of the indentations was 70 μm, more than 10 times the indentation depth (2-4 μm), sufficient to obtain accurate results for a Berkovich indenter (Ref. 2 in Supplementary Materials, Materials & Design 164, 107563, 2019).

From the load–depth hysteresis, E and H were calculated using the Oliver-Pharr method³.

$$H = \frac{P_{max}}{A(h_c)}$$

$$E_r = \frac{S\sqrt{\pi}}{2\beta\sqrt{A}}$$

$$\frac{1}{E_r} = \frac{1 - \nu^2}{E} + \frac{1 - \nu_i^2}{E_i}$$

Where P_{max} was directly obtained from the maximum load and A was the calibration area function of the indenter in contact depth h_c . E_r was defined as the equivalent elastic modulus. S was the slope of the unloading curve and β was a known dimensionless constant that depends on the geometry of the indenter (β of Berkovich = 1.034). E was converted from the calculation of E_r , which considered the deformation of both the indenter and sample. ν was Poisson's ratio of material. The elastic modulus and Poisson's ratio (E_i, ν_i) of the Berkovich indenter was 1141 GPa and 0.07. Poisson's ratio of samples was assumed to be 0.25.

Strain rate sensitivity test. A nanoindentation testing technique called "strain-rate jump tests" was utilized to measure the strain-rate sensitivity of materials, where a standard CSM method was adapted to perform several abrupt changes in the applied strain rate at defined indentation depths during one single indentation.⁴⁻⁷ For performing the strain-rate jumps on the samples, the indentation strain rate was kept constant up to an indentation depth of ~ 2000 nm. Afterward, changes in the strain rate were applied every 300 nm. Four different strain rates (from 0.12 to 0.004 s⁻¹) were used during a single indentation experiment with a 5 nm amplitude and 45 Hz oscillation.

The steady-state creep strain rate which can be empirically related with rupture time by Monkman-Grant equation (Ref. 8 in Supplementary Materials, Mechanical metallurgy, 3, 1976) is known to be strongly dependent on the applied stress σ , (absolute) temperature T , and grain size d :

$$\dot{\epsilon} = f \left(\frac{b}{d} \right)^p \left(\frac{\sigma}{G} \right)^n \exp \left(-\frac{Q}{RT} \right)$$

Where f is a material- and temperature-related factor, G is the shear modulus, b is the magnitude of Burgers vector, Q is the activation energy for creep, R is the gas constant, p is the inverse grain-size exponent, and n is the creep stress exponent. In this equation, the stress exponent, $n = (\partial \ln \dot{\epsilon}') / (\partial \ln \sigma)$, is often considered as a valuable indicator for the predominant creep mechanism. For each indentation test in this Berkovich indenter indentation, the mean stress σ and the strain rate $\dot{\epsilon}$ are often considered as,

$$\sigma \propto H = \frac{P_{max}}{\Psi h^2}$$

$$\dot{\epsilon} = \frac{1}{h} \frac{dh}{dt} = \frac{1}{2} \left(\frac{\dot{P}}{P} - \frac{\dot{H}}{H} \right) \approx \frac{1}{2} \frac{\dot{P}}{P}$$

Here Ψ is the constant related with tip geometry (e.g., 24.56 for the Berkovich

tip). And the strain rate sensitivity exponent (m , simply by $m=1/n$) at constant temperature can be described as (Ref. 9 in Supplementary Materials, Acta Materialia 54, 7, 2006; Ref. 10 in Supplementary Materials, Journal of Materials Research 19, 51, 2004),

$$m_{\text{nanoindentation}} = \frac{d \ln H}{d \ln \dot{\epsilon}_{\text{nanoindentation}}}$$

Then the strain rate sensitivity exponent m is obtained from the slope of that $\ln(H)/\ln(\dot{\epsilon})$ curve.

Fig. S18 | Digital photo of polished samples for nanoindentation. Black zone is sample, and transparent part is epoxy resin for cold inlay with sample fixed.

Fig. S19 | Schematic of specialized fluid cell with mineral oil to prevent the sulfide sample from being exposed to air during nanoindentation.

Fig. S20 | Nanoindentation strain-rate jump experiment by four different applied strain rates at corresponding displacement.

Fig. S21 | Mechanical characteristics of Se sample by nanoindentation test. (a-c) Displacement control mode for measurement of E and H by maintaining a constant strain rate of 0.05 s^{-1} . Using the Oliver-Pharr method, (a) the corresponding load–displacement curves and the calculated (b) E and (c) H. (d-f) Nanoindentation strain-rate jump experiment with four different applied strain rates. (d) Corresponding load–displacement curves, (e) the measured H, and (f) the calculated strain-rate sensitivity exponent m for Se. (g, h) Residual topography after nanoindentation experiments by microscopy. Scale bar: 100 μm for (g), 10 μm for (h).

Fig. S22 | Mechanical characteristics of Li_2Se sample by nanoindentation test. (a-c) Displacement control mode for measurement of E and H by maintaining a constant strain rate of 0.05 s^{-1} . Using the Oliver-Pharr method, (a) the corresponding load–displacement curves and the calculated (b) E and (c) H . (d-f) Nanoindentation strain-rate jump experiment with four different applied strain rates. (d) Corresponding load–displacement curves, (e) the measured H , and (f) the calculated strain-rate sensitivity exponent m for Li_2Se . (g, h) Residual topography after nanoindentation experiments by microscopy. Scale bar: 50 μm for (g), 10 μm for (h).

Fig. S23 | Mechanical characteristics of MSe sample by nanoindentation test. (a-c) Displacement control mode for measurement of E and H by maintaining a constant strain rate of 0.05 s^{-1} . Using the Oliver-Pharr method, (a) the corresponding load–displacement curves and the calculated (b) E and (c) H. (d-f) Nanoindentation strain-rate jump experiment with four different applied strain rates. (d) Corresponding load–displacement curves, (e) the measured H, and (f) the calculated strain-rate sensitivity exponent m for MSe. (g, h) Residual topography after nanoindentation experiments by microscopy. Scale bar: 100 μm for (g), 10 μm for (h).

Fig. S24 | Mechanical characteristics of Li_4MSe sample by nanoindentation test. (a-c) Displacement control mode for measurement of E and H by maintaining a constant strain rate of 0.05 s^{-1} . Using the Oliver-Pharr method, (a) the corresponding load–displacement curves and the calculated (b) E and (c) H . (d-f) Nanoindentation strain-rate jump experiment with four different applied strain rates. (d) Corresponding load–displacement curves, (e) the measured H , and (f) the calculated strain-rate sensitivity exponent m for Li_4MSe . (g, h) Residual topography after nanoindentation experiments by microscopy. Scale bar: 100 μm for (g), 10 μm for (h).

Fig. S25 | Mechanical characteristics of FeS sample by nanoindentation test. (a-c) Displacement control mode for measurement of E and H by maintaining a constant strain rate of 0.05 s^{-1} . Using the Oliver-Pharr method, (a) the corresponding load–displacement curves and the calculated (b) E and (c) H. (d-f) Nanoindentation strain-rate jump experiment with four different applied strain rates. (d) Corresponding load–displacement curves, (e) the measured H, and (f) the calculated strain-rate sensitivity exponent m for FeS. (g, h) Residual topography after nanoindentation experiments by microscopy. Scale bar: 100 μm for (g), 10 μm for (h).

Q3. It is unclear why the authors emphasize closure of the pores in Se cathode as “striking observation”. Upon lithiation there is a $\sim 98\%$ volume expansion of Se so the reduction of pores should not be a surprise.

A3. We thank the reviewer very much for the comments. We agree that the reduction of pores is not a surprise during the lithiation. However, we find that, even after the delithiation (i.e., after a full cycle of lithiation and delithiation), the closure of the pores in the Se cathode is still obvious. If no Se creep, SEM may show that the particles expand due to lithiation with pores shrinking during discharge and particles contract due to delithiation with re-enlarged pores during charging. By contrast, the electrode pores appear to further contract during charging (delithiation), according to the *in-situ* SEM observations (Fig. 3c-IV, V). This implies that the particles continue to deform into the pores even during charging. We believe this is a “striking observation” because it deviates with our instinctive expectation that the particles will shrink to their initial volume after delithiation with the pore reopened, instead of the observed further closure. It clearly

demonstrates that the particles should undergo some levels of inelastic deformation, mainly the Se/Li₂Se creep deformation.

Main text: “----- As shown in Fig. 3c-II, III, the pores undergo a distinct shrinkage with an evident enhancement in interparticle contact during discharge (lithiation), and the Se particles retain their structural integrity without being pulverized or broken. During subsequent charge (delithiation), Li-ion is extracted from Se particles, theoretically inducing Se retraction and pore enlargement. Supposedly, if there is no irreversible creep deformation, the pores volume should re-expand to 78% of the initial volume (Fig. S10), because the degree of lithiation of Se remains at 22% (calculated by the Columbic efficiency (CE)) at the end of charge (Fig. 3c-V). However, the actual porosity further contracts during charge (delithiation, Fig. 3c-IV, V), implying that the particles continue to deform into the pores due to Se creep. This phenomenon can be exclusively ascribed to the creep deformation induced by the internal stresses exerted on the Se particles by the rigid Mo₆Se₈ framework. We calculate the relative areas of 20 individual pores to substantiate our observations further. The results reveal that most pore sizes are much smaller than 78% of the initial state, and eight pores undergo further shrinkage during delithiation (Fig. 3d, Fig. S11), highlighting the pervasive nature of Se/Li₂Se creep within the electrode structure. This is a direct observation that Se creeps throughout the charging and discharging process, forming improved contact and releasing stress in the electrode.”

Q4. What is the “time-dependent hysteresis” that the authors mention on line 189?

A4. We thank the reviewer very much for the comments. The term “time-dependent hysteresis” in the sentence “Since the LCO and MSe are assumed to be linear elastic materials without any time-dependent hysteresis” mainly refers to the viscoelastic features of materials such as creep. To be more specific and avoid any ambiguities, we have replaced the “time-dependent hysteresis” term by the “creep behaviors”.

Q5. In supplementary materials the nano indentation figures are captioned as “Resulting hardness(modulus) in the nanoindentation strain-rate jump experiment.” What is the strain-rate jump experiment? The description of nano indentation experiments (albeit brief) states that they were done in CSM mode. Could the authors elaborate how the “strain rate jumps” were incorporated into the experiments and what was their purpose?

A5. We thank the reviewer very much for the comments. We apologized for missing the detail about the nanoindentation testing technique in previous supplementary materials. Verena Maier developed the “strain rate jumps” testing technique using a nanoindentation to measure the strain-rate sensitivity of materials (Ref. 7 in

supplementary materials, J. Mater. Res., 26, 11, 2011, *Nanoindentation strain-rate jump tests for determining the local strain-rate sensitivity in nanocrystalline Ni and ultrafine-grained Al*), where a standard CSM method was adapted to perform several abrupt changes in the applied strain rate at defined indentation depths during one single indentation. The results of the “strain rate jumps” tests have been compared to conventional constant strain-rate nanoindentation experiments, macroscopic compression tests, and finite element modeling simulations to demonstrate its accuracy and high efficiency in determine the strain-rate sensitivity of materials.

In specific, the steady-state creep strain rate can be empirically described as $\dot{\epsilon} = f \left(\frac{b}{d} \right)^p \left(\frac{\sigma}{G} \right)^n \exp \left(-\frac{Q}{RT} \right)$, (Ref. 8 in Supplementary Materials, Mechanical metallurgy, 3, 1976), where f is a material-and temperature-related factor, G is the shear modulus, b is the magnitude of Burgers vector, Q is the activation energy for creep, R is the gas constant, p is the inverse grain-size exponent, and n is the creep stress exponent, σ is the applied stress, T is (absolute) temperature, d is grain size. The indentation strain rate can be derived from the concept of true strain and can be estimated as $\dot{\epsilon} = \frac{1}{h} \frac{dh}{dt} = \frac{1}{2} \left(\frac{\dot{P}}{P} - \frac{\dot{H}}{H} \right) \approx \frac{1}{2} \frac{\dot{P}}{P}$. For each indentation test in this Berkovich indenter indentation, the mean stress σ can be estimated as $\sigma \propto H = \frac{P_{max}}{\psi h^2}$, or can be either assumed as the same as H or converted by Tabor’s empirical law, $\sigma = H/C$, where C is the constraint factor of depending on the material (Ref. D. Tabor: The Hardness of Metals (Oxford University Press, London, 1951)). Hence, the SRS can be deduced: $m_{nanoindentation} = \frac{d \ln H}{d \ln \dot{\epsilon}_{nanoindentation}}$. Then, the strain-rate sensitivity m is determined from the slope of that log-log plot of ln H/ln ($\dot{\epsilon}$).

The detailed description of the “strain rate jumps” testing technique using a nanoindentation has been added into the revised manuscript, as presented in our Answer #2 to the Question #2 from the reviewer.

Q6. Boundary conditions of the numerical simulations should be clearly explained (especially considering that a 3D problem was replaced with a 2D one).

A6. We thank the reviewer very much for the comments. In the simulations, the 2D plane strain assumption was used to simplify the 3D problem into the 2D one. The symmetric boundary conditions were applied to the surrounding boundaries of the 2D model to constrain the free deformation of the composite cathode.

To clarify the boundary conditions of the numerical simulations, we have added the following description into the revised Supplementary Materials – Section 1. Materials and Methods – Numerical Simulation.

“We build a two-dimensional **plane-strain** model consisting of two components—active material and electrolyte material, where lithiation only occurs in the active material. **The symmetric boundary conditions were applied to the surrounding boundaries of the 2D model to constrain the free deformation of the composite cathode (Fig. 4a).**”

Q7. The manuscript requires substantial re-write to improve the language.

A7. We thank the reviewer very much for pointing out this. We have polished the manuscript to improve the language.

REVIEWER COMMENTS

Reviewer #1 (Remarks to the Author):

The authors have carefully responded to my concerns, and I am generally satisfied with the authors' revisions. Therefore, I recommend that the work be published after addressing several minor concerns:

1. There are missing spaces between some numbers and units. Please revise it.
2. In Fig. 6, the font of the "°C" is inconsistent with others, please revise it and check the whole manuscript.
3. Please adjust the y-axis of the CE data to 90~100% for better readership.

Reviewer #2 (Remarks to the Author):

The authors have put efforts on carefully revising my comments to improve the quality of the manuscript. The following remaining questions and confusions still needs to be answered in order to have a consideration of a publication in Nature Communications.

1. When we approach "creep" behavior and concept, from the definition point, the authors have to demonstrate both visco-elastic and visco-plastic deformation of the designed materials. However, from the videos and the relevant experiments, the major conclusion can be made was the materials showed volume change and that volume change can introduce additional stress concentration at the particle/different grain junctions. Therefore, in order to state "creep"-type, the authors need to show clear experimental evidence that the designed materials showed both visco-elastic and visco-plastic deformation and responses. Otherwise, there are concerns that it is due to pure plastic deformation etc. instead of "creep".
2. Following by Question #1, do the authors have any clues on the time length scale that Se creeps and then stops creeping? When we compare Li metal, a very soft material, Li creep will be finished within one second. However, in the answers to original question #2, the authors stated that Se creep can occur for a relatively long time. Why and how? it is counterintuitive to me. Again, there is a high possibility and concern that it is due to the plastic deformation of Se instead of creep.
3. When the authors state "creep", was "creep" really originated from Se or was it from the formed CEI or both? Need explanation and comments.
4. At 100 MPa, there is severe Li creep happend. The authors need to comment on how Li creep would affect the results in the manuscript.
5. How large the particle size is? It seems to me from Figure S26, the particle size of Se is in the nano-meter range. This is far below the pore size and the size of nano-indenter. That means that when the authors performed the nano-indentation, the indenter will touch the pore areas anyway. This will lead to a dramatic reduction in the Young's modulus. However, the authors still get a slightly higher or same level of Young's modulus (Table S6) for dense Se material, for example. This is counter-intuitive to me as well.

Reviewer #3 (Remarks to the Author):

While the softening of the Li₂Se phase contradicts its higher melting temperature, the authors supported the analysis by in-house nanoindentation experiments demonstrating such softening. The authors responded to my questions. This work is worth publishing.

Response to Reviewer #2

Comment: The authors have put efforts on carefully revising my comments to improve the quality of the manuscript. The following remaining questions and confusions still needs to be answered in order to have a consideration of a publication in Nature Communications.

Reply: We highly appreciate the reviewer's constructive suggestions, which help us further improve the quality of our manuscript. All your concerns are addressed point-to-point.

Q1. *When we approach "creep" behavior and concept, from the definition point, the authors have to demonstrate both visco-elastic and visco-plastic deformation of the designed materials. However, from the videos and the relevant experiments, the major conclusion can be made was the materials showed volume change and that volume change can introduce additional stress concentration at the particle/different grain junctions. Therefore, in order to state "creep"-type, the authors need to show clear experimental evidence that the designed materials showed both visco-elastic and visco-plastic deformation and responses. Otherwise, there are concerns that it is due to pure plastic deformation etc. instead of "creep".*

A1. We thank the reviewer for the comments. We conducted a series of nanoindentation experiments to demonstrate that the designed materials have both visco-elastic and visco-plastic deformation (**Fig. S27a**). In these measurements, we held the peak load of nanoindentations for an extended dwell time between loading and unloading. If the deformation of the designed material is time-independent (e.g., i.e., no viscosity), the indentation depth should remain constant during the dwell, while if the deformation is time-dependent (viscosity), the indentation depth will increase during the dwell. As shown in **Fig. S27b**, the indentation depth increases continuously with dwell time, indicating *the viscosity of the design material*.

Therefore, the material can be categorized into (1) viscoelasticity & no plasticity (e.g., rubbers or hydrogels), (2) elasticity & viscoplasticity (e.g., clays or concretes), or (3) viscoelasticity & viscoplasticity (e.g., polycrystalline metals). As shown in **Fig. S28**, the indentation depth reverts after unloading from h_2 to h_3 (Δh_1), leaving the non-reversible part (Δh_2) due to the plastic deformation. It demonstrates that *the designed material exhibits viscoplasticity*, excluding the category (1).

Next, we demonstrate that, in addition to the viscoplasticity, the designed material also has viscoelasticity occurring under a load without yielding the material. A.H.W. Ngan et al. suggest a nanoindentation method to virtualize the viscoelasticity of materials: the indentation load was first rapidly ramped up to a high peak value, followed by a slow unloading (Ref. *J. Mater. Res.*, 17, 10, 2002). Since no viscoplasticity can occur during the elastic unloading, the viscoelasticity of materials can be easily distinguished from the unloading curve as the "nose".

Unfortunately, we could not perform this experiment because our nanoindenter (G200 XP, Keysight) does not allow the artificial control of the unloading rate to fit the slow unloading reported in Ref.

However, our *in-situ* observation in **Fig. 3c** indicates that the designed material presents a similar viscoelastic behavior as reported in Ref. Initially, the Se particles tend to expand due to lithiation (discharge), leading to noticeable pore shrinkage, as shown in Fig. 3c-I, II, and III. During subsequent delithiation (charge, Fig. 3c-IV, V), the extraction of Li from Se particles should have caused the particles to shrink while the stress is being released. However, the particles continue to expand (Fig. 3c-IV and V), indicating they are still experiencing the viscous deformation associated with the lithiation. It is well analogous to the indentation depth increase during unloading (i.e., “nose”) reported by A.H.W. Ngan et al. Since the particles are in the elastic unloading state, the viscoplasticity should not participate in the viscous deformation; thus, it is attributed to the viscoelasticity of the material.

Fig. S27 | The constant load and hold method for creep measurements of Se material. The loading rate is 6.67 mN/s and the holding times is 100 s. (a) The load on the sample. (b) The displacement into surface curves versus time on the sample.

Fig. S28 | Load-displacement curves of Se sample at different loading rates and maximum loads (hold at the max load for 100 s).

Fig.R1 | Load versus displacement curves of polypropylene at different unloading rates. All three experiments had the same loading rate to the same peak load and the same holding time before unload (Ref. J. Mater. Res., 17, 10, 2002). [Ngan, A.H.W., Tang, B. Viscoelastic effects during unloading in depth-sensing indentation. *Journal of Materials Research* 17, 2604–2610 (2002), reproduced with permission from SNCSC.]

Fig. 3. In-situ observation of Se creep evolution during galvanostatic discharging-charging. (c)
The morphology of the Se-60MSe cathode varies with different states of charge.

To clarify the above comments, Fig. S27, Fig. S28, and the additional experimental details, as well as the following discussions, have been added to the revised Main Text and Supplement Materials:

Main Text:

“-----In addition, the nanoindentation test holding peak load for over 100 s of Se material further indicates that the Se is prone to continual creep (Fig. S27, S28). -----”

Supplement Materials:

“-----The constant load tests. Tests followed a typical chronology: Indenter loads at a constant loading rate until the given peak load, holds at the peak load (100 mN and 200 mN) for an extended dwell time (100 s), then partially unloads to 10% of the peak load and holds for a second dwell time (60 s) to measure the drift-rate, and finally completely withdraws from the sample.”

“As illustrated in this constant load test by nanoindentation, the indentation depth increases continuously during the long-term dwell time, indicating the viscosity feature of Se material.”

“After unloading, the indentation depth reverts from h_2 to h_3 (Δh_1), leaving the non-reversible part (Δh_2) due to the plastic deformation, which well further demonstrates that the Se material exhibits viscoplasticity.”

Q2. Following by Question #1, do the authors have any clues on the time length scale that Se creeps and then stops creeping? When we compare Li metal, a very soft material, Li creep will be finished within one second. However, in the answers to original question #2, the authors stated that Se creep can occur for a relatively long time. Why and how? it is counterintuitive to me. Again, there is a high possibility and concern that it is due to the plastic deformation of Se instead of creep.

A2. We thank the reviewer for the comments. According to the definition of creep, as long as the stress is applied to a creepable material, it will continue to creep over time until cracks and pores form within the material (**Fig. R2**). William S. LePage Group performed the creep deformation of Li foil over **15 h** by the standard uniaxial tensile test (Ref. J. Electrochem. Soc. 166, 2, 2019). Li deformation begins with transient creep and subsequently develops to prolonged steady-state creep (**15 h**, **Fig. R3a**), demonstrating that Li creep is a continuous process that occurs over time rather than a transient one with a limited duration.

The complexity of the cathode microstructure makes it challenging to quantify the internal stress applied to the Se particles accurately and, therefore, the associated creep time. However, the nanoindentation for the Se material in **Fig. S27** and **S28** (see above answers) indicates that the Se material experiences steady-state creep when holding the indentation loading, and this process lasts for over a hundred seconds. Moreover, the *in-situ* observation in **Fig. 3c** indicates the Se particles experience creep upon the entire delithiation, lasting over 7 hours. Last, the continuous increase in battery capacity during the first 480 h (**Fig.5a** in the main text) also indicates that the Se particles creep across multiple cycles because the cyclic expansion of Se particles in each cycle build up the internal stress to drive the creep, as illustrated in our numerical simulation in **Fig.4f**.

Fig. R2 | Creep strain curve and the various stages of creep behavior of materials.

Fig. R3 | Strain-rate dependent deformation measurements of Li foil by conventional uniaxial creep test with a digital image correlation (DIC). (a) The creep curves of true strain versus time at 298 K. (b). The mechanical responses of Li foil at 298 K are shown as true stress against true strain (c) A sequence of axial strain maps from DIC. (d) The true strain rate is plotted against the true stress along with the creep exponent (Ref. William S. LePage et al., J. Electrochem. Soc. 166, 2, 2019).

Fig. 3. *In-situ* observation of Se creep evolution during galvanostatic discharging-charging. (c) The morphology of the Se-60MSe cathode varies with different states of charge.

Fig. 5. (a) The cycling performance of CT-ASS Se-60MSe cathode at 0.05 C (1C = 0.96 mA/cm²) and the corresponding capacity contributed by Se.

Fig. 4. Numerical analysis of the creep behavior of the all-solid-state cathodes. (f) The equivalent stress and creep strain in the CT-ASS Se-MSe cathode at different cycles.

Q3. When the authors state "creep", was "creep" really originated from Se or was it from the formed CEI or both? Need explanation and comments.

A3. We thank the reviewer for the comments. As depicted in the schematic diagram (**Fig. R4**), the Se-60MSe cathode and the solid electrolyte LPSC are only in touch in the intermediate layer (Part 1). It is unlikely that LPSC will be oxidized to form CEI on the surface of Se particles. To confirm this, we investigated the differential capacity dQ/dV curves (**Fig. R5**) of the Se-60MSe|LPSC|InLi battery and found no characteristic peak of the decomposition voltage of LPSC in the operating voltage range of the Se-60MSe cathode.

Moreover, the Se-60MSe cathode itself only contains Se and MSe particles that do not react with each other, which is supported by the fact that the ball-milling mixed Se-60MSe powder still retains the Se and MSe phases with no generation of new phases. (**Fig. S7a**).

Lastly, it is well-accepted that the thickness of CEI is approximately 10 nm. It is difficult for this nanoscale interphase to alter the mechanical behaviors of the microscale active particles and the electrode.

Fig. R4 | The structure diagram of the Se-60MSe|LPSC|InLi all-solid-state batteries.

Fig. R5 | Comparison of the differential capacity dQ/dV curve of (a) the Se-60MSe|LPSC|InLi battery and (b) the oxidation and reduction peaks of LPSC from Ref. (Nature Materials, 19, 428-435, 2020).

Fig. S7 | (a) XRD patterns of Se powder, Mo_6Se_8 powder, and the composite Se-MSe powder obtained by ball milling the Se and Mo_6Se_8 (300 r/min, 3 h).

To clarify those comments, Fig. S7a, as well as the following discussions, have been added to the revised Supplement Materials:

“-----As illustrated by the XRD patterns, there are no new phases generated after the ball-milling mixing process, which suggests no reaction between Se and MSe, and the constructed Se-60MSe cathode remains a stable phase structure.”

Q4. At 100 MPa, there is severe Li creep happened. The authors need to comment on how Li creep would affect the results in the manuscript.

A4. We thank the reviewer for the comments. We elaborate on this comment from the following aspects:

- ① In this work, Indium (In) foil (Φ 10 mm, thickness 0.1 mm) and Li foil (Φ 9 mm, thickness 0.08 mm) are cold pressed into $\text{In}_{1.3}\text{Li}$ alloy (0.6 V vs Li/Li⁺) and used as anode (instead of metallic Li). The hardness of lithiated InLi alloy ($E \sim 46$ GPa, $H \sim 1.8$ GPa, $T_m \sim 600$ °C, Ref. ACS Energy Lett. 9, 2, 578–585, 2024) increase with increasing Li content. As shown in previous studies (Materials Science and Engineering, 93, 83-92, 1987), InLi alloy is brittle at low temperature without viscosity below about 200°C. Furthermore, in contrast with many other intermetallic compounds, LiIn alloy don't experience the defect softening phenomenon even at high homologous temperatures even at temperatures up to 450 °C ($\sim 0.8 T_m$), even though Indium ($E = 12.7$ GPa, $T_m = 156.6$ °C, Ref. Journal of Materials Research 36, 2444–2455, 2021, Ref. 17 in Supplementary Materials) is similar to Li and prone to creep and stress relaxation.

- ② Our work compares the cycling performance and morphological evolution between the creepable Se-MSe electrode and the non-creepable FeS-MSe electrode pairing with the same $\text{In}_{1.3}\text{Li}$ alloy anode. (**Fig. 5** in the main text). The improved interparticle contact and less fractured particles of Se-MSe cathode over FeS-MSe cathode indicate that the optimized performance is primarily due to Se creep. Therefore, the $\text{In}_{1.3}\text{Li}$ alloy anode should not affect our results in the manuscript.
- ③ In addition, we conducted a group of control experiments during the revision. We fabricated the composite cathode with the small-volume-variable LiCoO_2 (70wt% LCO+30wt% LPSC, $\Delta V_{\text{LCO}} \sim 2\%$) and the zero-strain $\text{Li}_4\text{Ti}_5\text{O}_{12}$ (40wt% LTO+60wt% LPSC+10wt% Super P, $\Delta V_{\text{LTO}} \sim 0.2\%$) and paired them with an $\text{In}_{1.3}\text{Li}$ anode. As reported in the manuscript, none display increasing capacity during cycling (**Fig. R6**). It suggests that the continual capacity increase in Se-60MSe|LPSC|InLi batteries is mainly derived from the Se creep.
- ④ Lastly, even though the practical ASSBs might eventually use Li foil as the anode (**Fig. R7**), the creep of Li foil only improves the contact condition between the Li and SSE (Ref. ACS Appl. Mater. Interfaces, 13, 22, 26533–26541, 2021), while not affects the microstructural evolution within the cathode (as proved in 1-3). In this context, we believe the electrochemical-mechanical stability of the ASSBs could be further improved.

Fig. R6 | The electrochemical performance of LCO|LPSC|InLi and LTO|LPSC|InLi all-solid-state batteries. (a), (b) LCO|LPSC|InLi; (c), (d) LTO|LPSC|InLi.

Fig. R7 | The structure schematic of all-solid-state batteries with Li anode.

To support our claim further of the creep evolution of the porous Se-MSe cathode, we have added the the following discussions to the revised Main Text and made an explanation in a supplementary note in Section 2. Supplementary Text to eliminate the interference of the InLi alloy anode and Li anode.

Main text:

“----- It should be noted that our used $\text{In}_{1.3}\text{Li}$ alloy anode is brittle at low temperature without viscosity below about 200°C , though Indium is similar to Li and prone to creep and stress relaxation (detailed information shown in Supplementary Note). Thus, the $\text{In}_{1.3}\text{Li}$ anode don't affect the creep evolutionary of the Se-60MSe cathode.”

Supplement Material:

“Supplementary Note

The hardness of lithiated InLi alloy ($E \sim 46$ GPa, $H \sim 1.8$ GPa, $T_m \sim 600^\circ\text{C}$) increase with increasing Li content¹⁶. And it should be noted that InLi alloy is brittle at low temperature without viscosity below about 200°C ¹⁷. Furthermore, in contrast with many other intermetallic compounds, LiIn alloy don't experience the defect softening phenomenon even at high homologous temperatures even at temperatures up to 450°C ($\sim 0.8 T_m$), even though Indium ($E = 12.7$ GPa, $T_m = 156.6^\circ\text{C}$) is similar to Li and prone to creep and stress relaxation¹⁸.

In addition, even though the practical ASSBs might eventually use Li foil as the anode, the creep of Li foil only improves the contact condition between the Li and SSE, while not affects the microstructural evolution within the cathode.

”

Q5. How large the particle size is? It seems to me from Figure S26, the particle size of Se is in the nano-meter range. This is far below the pore size and the size of nano-indenter. That means that when the authors performed the nano-indentation, the indenter will touch the pore areas anyway. This will lead to a dramatic reduction in the Young's modulus. However, the authors still get a slightly higher or same

level of Young's modulus (Table S6) for dense Se material, for example. This is counter-intuitive to me as well.

A5. We thank the reviewer for the comments. Since we aim to characterize the intrinsic mechanical properties of Se materials (instead of the composite cathode), we first prepare a Se pellet sample as densely as possible. We cold-press the Se particles (1-3 μm , **Fig. S29**) into a dense pellet using a high uniaxial compressive load of 10 t (~ 1250 MPa) at room temperature and then polish the surface for nanoindentation. The surface morphology and inner microstructure of the dense Se pellet are imaged using SEM (**Fig. S19a-c**). The Se pellet has a dense and smooth surface, and the particles have intimate contact without invisible pores and cracks (**Fig. S19d-f**), consistent with its high relative density of 95.65% (measured by Archimedes method). The Young's modulus and hardness measured from such a dense pellet should be close to the material's intrinsic properties. Meanwhile, each nanoindentation test was conducted at least ten points to guarantee the statistical reliability of the results.

Fig. S19 | The surface morphology of cold-pressed Se samples used for nanoindentation. (a-c) The polished surface and (d-f) inner microstructure.

To clarify the above comments, Fig S19, as well as the following discussions, have been added to the revised Main Text and Supplement Materials:

Main Text:

“-----The parameters for numerical analysis are listed in Table S6.”

“-----The samples used for nanoindentation tests have a dense and smooth surface, and the particles have intimate contact without invisible pores and cracks. Thus, the mechanical properties measured from such dense pellets should be close to the material's intrinsic properties (Fig. S19). -----”

Supplementary Materials:

“-----The Se pellet has a much dense contact in particles without invisible pores and cracks, consistent with its high relative density of 95.65% (measured by Archimedes method). The Young's modulus and hardness measured from such dense pellet should be close to the material's intrinsic properties.”

REVIEWERS' COMMENTS

Reviewer #2 (Remarks to the Author):

The authors have provided a careful revision and resolved my comments. Thus, approves for a publication in Nature Communications.